# Pay attention to your loss: understanding misconceptions about 1-Lipschitz neural networks

**Louis Béthune,** [†]
IRIT, Université Paul-Sabatier
Toulouse, France

**Thibaut Boissin,** [†]
IRT Saint-Exupéry
Toulouse, France

**Mathieu Serrurier**
IRIT, Université Paul-Sabatier
Toulouse, France

**Franck Mamalet**
IRT Saint-Exupéry
Toulouse, France

**Corentin Friedrich**
IRT Saint-Exupéry
Toulouse, France

**Alberto González-Sanz**
IMT, Université Paul-Sabatier
Toulouse, France

## Abstract

Lipschitz constrained networks have gathered considerable attention in the deep learning community, with usages ranging from Wasserstein distance estimation to the training of certifiably robust classifiers. However they remain commonly considered as less accurate, and their properties in learning are still not fully understood. In this paper we clarify the matter: when it comes to classification 1-Lipschitz neural networks enjoy several advantages over their unconstrained counterpart. First, we show that these networks are as accurate as classical ones, and can fit arbitrarily difficult boundaries. Then, relying on a robustness metric that reflects operational needs we characterize the most robust classifier: the WGAN discriminator. Next, we show that 1-Lipschitz neural networks generalize well under milder assumptions. Finally, we show that hyper-parameters of the loss are crucial for controlling the accuracy-robustness trade-off. We conclude that they exhibit appealing properties to pave the way toward provably accurate, and provably robust neural networks.

## 1 Introduction

1-Lipschitz neural networks have drawn great attention in the last decade, with motivation ranging from adversarial robustness to Wasserstein distance computation. In the following, we denote by **LipNet1** the class of 1-Lipschitz neural networks, by **AllNet** the class of neural networks without constraints on their Lipschitz constant, i.e conventional neural networks.

Roughly speaking, the Lipschitz constant of neural networks quantifies how much their outputs can change when inputs are perturbed. When this constant is high, as it is often the case for neural networks of AllNet , they become vulnerable to adversarial attacks (see [1, 2] and references therein): a carefully chosen small noise added to the inputs, usually imperceptible, can change the class prediction. One possible defense against adversarial attacks is to constrain the network to be 1-Lipschitz (in LipNet1 ) [3], which provides provable robustness guarantees, together with an improvement of generalization [4] and interpretability of the model [5]. LipNet1 networks are also used to estimate Wasserstein distance, thanks to Kantorovich-Rubinstein duality in the seminal work of WGAN [6].

Despite their competitiveness with networks of AllNet on medium scale problems [7, 8], they still suffer from misconceptions. A belief commonly invoked against networks of LipNet1 is that they are less expressive: "Lipschitz-based approaches suffer from some representational limitations that may

36th Conference on Neural Information Processing Systems (NeurIPS 2022).

prevent them from achieving higher levels of performance and being applicable to more complicated problems" [9].

Although this claim seems rational at first glance, the link between Lipschitz constant and expressiveness is not trivial. While there is an obvious lack of expressiveness for regression tasks, this intuition fades when it comes to classification. Indeed, every AllNet network $g : \mathbb{R}^n \to \mathbb{R}^K$ is $L$-Lipschitz for some (generally unknown) $L > 0$. Then $f = \frac{1}{L}g$ is a 1-Lipschitz neural network with the same decision boundary, since prediction $\arg\max_k g_k$ is invariant by positive rescaling of the logits. In particular, $f$ has the same accuracy and also the same robustness to adversarial attacks as $g$. We illustrate this empirically by **training a LipNet1 network until it reaches 99.96% accuracy on CIFAR-100 with random labels** (see Appendix I).

We demonstrate that LipNet1 networks are theoretically better grounded than AllNet networks when it comes to classification, through our threefold contribution on Expressiveness (Section 3), Robustness (Section 4) and Generalization (Section 5).

**First, in Section 3** we confirm that LipNet1 are as expressive as AllNet networks for classification, and can learn arbitrary complex decision boundary. We show that hyper-parameters of the loss are of crucial importance, and control the ability to fit properly the train set.

**Then, in Section 4** we show that accuracy and robustness are often antipodal objectives. We characterize the robustness of the highest accuracy LipNet1 classifier: it is achieved by the Signed Distance Function (Definition 6 in Appendix A). We also characterize the classifier of highest certifiable robustness, and we show it corresponds to the dual potential of Wasserstein-1 distance (i.e the discriminator of a WGAN [6]).

**Finally, in Section 5** we show that LipNet1 benefit from several generalization guarantees. They are consistent estimators: contrary to AllNet , we prove that their train loss will converge to test loss as the size of the train set increases. Moreover, we show that LipNet1 classifiers with margin are PAC-learnable [10]: it provides bounds on the number of train examples required to reach a targeted test accuracy. Interestingly, this bound is independent of the architecture size, which allows to train enormous LipNet1 networks without risking overfitting.

## 2 Notations and experimental setting

The core of the paper mainly deal with binary classification over $\mathbb{R}^n$ with label set $\mathcal{Y} = \{-1, +1\}$. Let $(X, Y)$ be a random variable taking values on $\mathcal{X} \times \mathcal{Y}$, where $\mathcal{X} \subset \mathbb{R}^n$ is assumed to be a compact set. Such a pair follows the joint distribution $\mathbb{P}_{XY}$, defined on the space of probability measures $\mathcal{P}(\mathcal{X} \times \mathcal{Y})$. The marginal distribution of $X$ is denoted by $\mathbb{P}_X \in \mathcal{P}(\mathcal{X})$ and its support by $\operatorname{supp} \mathbb{P}_X$. We suppose the observation of a sample $(x_1, y_1), \ldots, (x_p, y_p)$ i.i.d. with common law $\mathbb{P}_{XY}$, and the goal is to learn a classifier $c : \mathcal{X} \to \mathcal{Y}$ modeling the optimal Bayes classifier $\arg\max_{y \in \mathcal{Y}} \mathbb{P}_{Y|X}(y|x)$. $P$ (resp. $Q$) denotes the input distribution of label $+1$ (resp. $-1$).

The Lipschitz constant $\operatorname{Lip}(f)$ of a function $f : \mathbb{R}^n \to \mathbb{R}^K$ is defined as the smallest $L \geq 0$ such that for all $x, z \in \mathbb{R}^n$ we have $\|f(x) - f(z)\| \leq L\|x - z\|$. In the rest of the paper, we focus on euclidean norm $\|\cdot\|$ for vectors and spectral norm $\|\cdot\|_2$ for matrices. The set of $L$-Lipschitz functions over $\mathcal{X} \subset \mathbb{R}^n$ with image in $\mathbb{R}^K$ is denoted $\operatorname{Lip}_L(\mathcal{X}, \mathbb{R}^K)$.

**Definition 1** (Class of AllNet networks)
*AllNet denotes the set of unconstrained neural networks. It includes any feed-forward network of fixed depth (without recurrent mechanisms) using affine layers (including convolutions and batch normalization) with weight matrices $W_1, W_2, \ldots W_d$ and Lipschitz activation function $\sigma$ (such as ReLU, sigmoid, tanh, etc). No constraint is enforced on their Lipschitz constant during training.*

**Definition 2** (Class of LipNet1 networks)
*LipNet1 denotes the set of feed-forward neural networks $f$ defined as in Theorem 3 of Anil et al. [11]: $\|W_1\|_{2\to\infty} \leq 1$ (see [12] for details on the mixed norm $\|\cdot\|_{2\to\infty}$) and $\|W_i\|_\infty \leq 1$ for $i \geq 2$, and GroupSort2 activation function. They fulfill $\operatorname{Lip}(f) \leq 1$.*

**Remark.** *AllNet networks benefit from universal approximation theorem in $C(\mathcal{X}, \mathbb{R}^K)$, a classical result of literature [13]. LipNet1 networks also benefit from an universal approximation theorem in $\operatorname{Lip}_1(\mathcal{X}, \mathbb{R})$ with respect to uniform convergence [11]. Note that $\operatorname{Lip}_L(\mathcal{X}, \mathbb{R}^K) = \{Lf \mid f \in \operatorname{Lip}_1(\mathcal{X}, \mathbb{R}^K)\}$ so LipNet1 can be used to approximate functions in $\operatorname{Lip}_L(\mathcal{X}, \mathbb{R}^K)$.*

In practice authors of [11] noticed that using orthogonal weight matrices (i.e $W_i^T W = I$) yielded the best results. All our experiments use the Deel.Lip[1] library [8], following ideas of [11]. The networks use 1) orthogonal weight matrices and 2) GroupSort2 activations [11]. Orthogonalization is enforced using Spectral normalization [14] and Björck algorithm [15]. These networks belong to LipNet1 by construction (see Appendix D for our choice of architecture and relevant related work).

LipNet1 networks provide robustness radius certificates against adversarial attacks [16]. Computing these certificates is straightforward and does not increase runtime, contrary to methods based on bounding boxes or abstract interpretation [17, 18, 19, 20]. There is no need for adversarial training [21] that fails to produce guarantees, or for randomized smoothing [22] which is costly.

Confusingly, any network of AllNet has a finite Lipschitz constant, but computing it is NP-hard [23]. Only a loose upper bound can be cheaply estimated: $\text{Lip}(f) \leq \text{Lip}(\sigma)^d \Pi_{i=1}^d \|W_i\|_2$ using the property that $\text{Lip}(f_d \circ f_{d-1} \circ \ldots \circ f_1) \leq \Pi_{i=1}^d \text{Lip}(f_i)$. In practice, this bound is often too high to provide meaningful certificates and besides, AllNet networks have usually very small robustness radius [1].

**Definition 3** (Adversarial Attack)
*For any classifier $c : \mathcal{X} \to \mathcal{Y}$, any $x \in \mathbb{R}^n$, consider the following optimization problem:*

$$\epsilon = \min_{\delta \in \mathbb{R}^n} \|\delta\| \text{ such that } c(x + \delta) \neq c(x). \tag{1}$$

*$\delta$ is an adversarial attack, $x + \delta$ is an adversarial example, and $\epsilon$ is the robustness radius of $x$.*

**Property 1** (Local Robustness Certificates [16]). *For any $f \in$ LipNet1 the robustness radius $\epsilon$ of binary classifier $\text{sign} \circ f$ at example $x$ verifies $\epsilon \geq |f(x)|$.*

**Losses:** The Binary Cross-Entropy (BCE) loss (also called logloss) is among the most popular choices of loss within the deep learning community. Let $f : \mathbb{R}^n \to \mathbb{R}$ a neural network. For an example $x \in \mathbb{R}^n$ with label $y \in \mathcal{Y}$, and $\sigma(x) = \frac{1}{1+\exp(-x)}$ the logistic function mapping logits to probabilities, the BCE is written $\mathcal{L}_\tau^{bce}(f(x), y) = -\log \sigma(y\tau f(x))$, with temperature scaling parameter $\tau > 0$. This hyper-parameter of the loss defaults to $\tau = 1$ in most frameworks such as Tensorflow or Pytorch. Note that $\mathcal{L}_\tau^{bce}(f(x), y) = \mathcal{L}_1^{bce}(\tau f(x), y)$ so we can equivalently tune $\tau$ or the Lipschitz constant $L$. We show in Section 5.1 that **for LipNet1 the temperature $\tau$ allow to control the generalization gap**. We also consider the Hinge loss $\mathcal{L}_m^H(f(x), y) = \max(0, m - yf(x))$ with margin $m > 0$, as used in [3] for LipNet1 networks training.

We focus on binary classification for readability and clarity purposes; however, we prove in Appendices A.2 and E that **the following theoretical results generalize to the multi-class case**, as done in experiments. The proofs of all propositions can be found in the appendix.

# 3  1-Lipschitz classifiers are expressive

In this section, we show that LipNet1 are as powerful as any other classifier, like their unconstrained counterpart. In particular, when classes are separable they can achieve 100% accuracy.

## 3.1  Boundary decision fitting

**Proposition 1. Lipschitz Binary classification.** *For any binary classifier $c : \mathcal{X} \to \mathcal{Y}$ with closed pre-images ($c^{-1}(\{y\})$ is a closed set) there exists a 1-Lipschitz function $f : \mathbb{R}^n \to \mathbb{R}$ such that $\text{sign}(f(x)) = c(x)$ on $\mathcal{X}$ and such that $\|\nabla_x f\| = 1$ almost everywhere (w.r.t Lebesgue measure).*

The level-sets of a $\text{Lip}_1(\mathcal{X}, \mathbb{R}^K)$ functions (and especially the decision boundary) can be arbitrarily complex: restraining classifiers to $\text{Lip}_1(\mathcal{X}, \mathbb{R})$ does not affect the classification power.

**Definition 4** ($\epsilon$-**separated distributions**)
*Distributions $P$ and $Q$ are $\epsilon$-separated if the distance between supp $P$ and supp $Q$ exceeds $\epsilon > 0$.*

**Corollary 1. Separable classes implies zero error**. *If $P$ and $Q$ are $\epsilon$-separated, then there exists a network $f \in$ LipNet1 such that **error** $E(\text{sign} \circ f) := \mathbb{E}_{(x,y) \sim \mathbb{P}_{XY}}[\mathbb{1}\{\text{sign}(f(x)) \neq y\}] = 0$.*

---

[1]`https://github.com/deel-ai/deel-lip` distributed under MIT License (MIT).

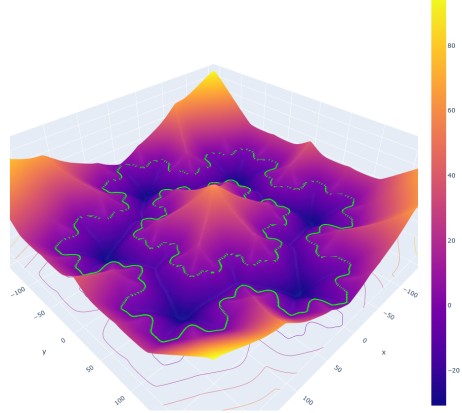

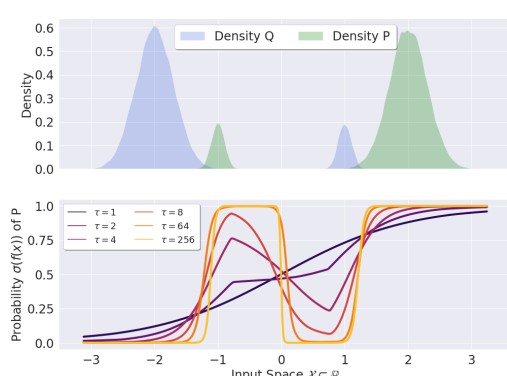

(a) **Complex Decision Boundary** $\partial$. We chose $\partial$ as the fourth iteration of Von Koch Snowflake. We chose $P$ as the interior ring, while the center and the exterior correspond to $Q$. We train a LipNet1 network with MSE to fit the SDF (Definition 6 in Appendix) ground truth (160 000 pixels), until MAE is inferior to 1. It proves empirically that LipNet1 networks can handle very sharp (almost fractal) decision boundary.

(b) **Importance of** $\tau$ **in BCE**. We train a LipNet1 network with BCE and different values for $\tau$. We chose a toy example where $P$ and $Q$ are Gaussian mixtures with two modes of weights $0.9$ and $0.1$. We highlight the different shapes of the minimizer $\sigma \circ f$ as function of $\tau$. **High values of** $\tau$ **leads to better fitting, whereas for lower** $\tau$ **the small weights Gaussian of the mixture are treated as noise and ignored**.

Figure 1

The class of LipNet1 networks does not suffer from bias for classification tasks. Some empirical studies show that indeed most datasets classes are separable [24] such as CIFAR10 or MNIST. Furthermore, even if the classes are not separable, functions of LipNet1 can nonetheless approximate the optimal Bayes classifier. Lipschitz constraint is not a constraint on the shape of the boundary (Figure 1a), but on the slope of the landscape of $f$.

## 3.2   Understanding why LipNet1 are often perceived as not expressive

LipNet1 networks cannot reach zero loss with BCE: this may explain why they are perceived as not expressive enough. Yet the minimizer of BCE exists and is well defined.

**Proposition 2.  BCE minimization for 1-Lipschitz functions**. *Let $\mathcal{X} \subset \mathbb{R}^n$ be a compact and $\tau > 0$. Then the infimum in Equation 2 is a minimum, denoted $f^\tau \in Lip_1(\mathcal{X}, \mathbb{R})$:*

$$f^\tau \in \arg \inf_{f \in Lip_1(\mathcal{X}, \mathbb{R})} \mathbb{E}_{(x,y) \sim \mathbb{P}_{XY}} [\mathcal{L}_\tau^{bce}(f(x), y)]. \tag{2}$$

*Moreover, the LipNet1 networks will not suffer of vanishing gradient of the loss (see Appendix F).*

Machine learning practitioners are mostly interested in maximizing accuracy.However, the minimizer of BCE is not necessarily a minimizer of the error (see Figure 1b). Yet, BCE is notoriously a differentiable proxy of the error $E(\text{sign} \circ f)$, and as $\tau \to \infty$ we get asymptotically closer to maximum empirical accuracy. Bigger value for $\tau$ might ultimately lead to overfitting, playing the same role as the Lipschitz constant $L$ (see Figure 1b).

**The implicit parameter $\tau = 1$ of the loss is partially responsible of the poor accuracy of LipNet1 networks in literature**, and not by any means the hypothesis space LipNet1 itself. This can be observed in practice : when temperature $\tau$ (resp. margin $m$) of cross-entropy (resp. hinge loss) is correctly adjusted a small LipNet1 CNN can reach a competitive **88.2% validation accuracy on the CIFAR-10 dataset** (results synthetized and discussed in Figure 3) *without* residual connections, batch normalization or dropout. Conversely, AllNet networks are roughly equivalent to learning a LipNet1 network with $\tau \to \infty$: without regularization or data augmentation, such a network can always reach 100% train accuracy without generalization guarantees.

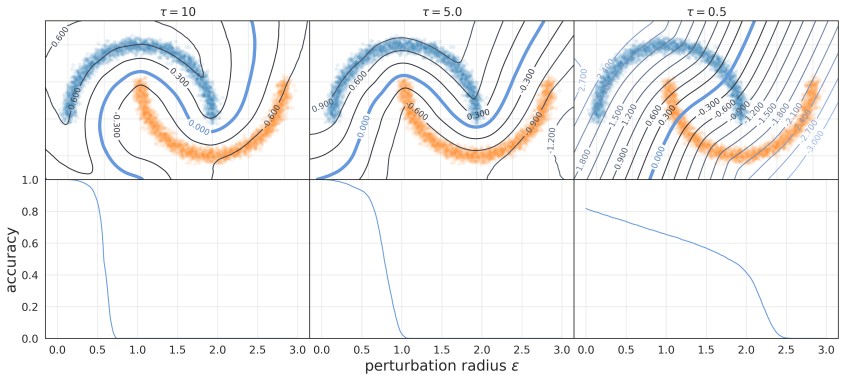

Figure 2: **Accuracy-robustness tradeoff:** Each network is optimal with respect to a certain criterion. The leftmost network is the most accurate at robustness radius $\epsilon \leq 0.3$, the rightmost maximizes the MCR at the cost of low clean accuracy. The center network corresponds to a compromise.

# 4 1-Lipschitz classifiers are certifiably robust

**Is there a trade-off between accuracy and robustness?** Although the existence of a trade-off between accuracy and robustness is commonly admitted, some works argue that "Robustness is not inherently at odds with accuracy"[24]. We propose a unified consideration by stating that for a given train accuracy, robustness can be maximized up to a certain point, but allowing a lower train accuracy helps achieving a higher robustness. Finally one must keep in mind that this trade-off lives in the shade of generalization (see Section 5).

## 4.1 Improving the robustness of the maximally accurate classifier

The Signed Distance Function [25] (SDF) (see Definition 6 in Appendix A) associated to the frontier $\partial$ of Bayes classifier $b$ is the 1-Lipschitz function that provides the largest certificates among the classifiers of maximum accuracy. Moreover, those certificates are exactly equal to the distance of adversarial samples. Iterative gradient based attacks (see [26] and references therein) can succeed in one step: far from being a weakness, this may improve the interpretability of the model [27, 28, 29].

**Corollary 2.** *For the SDF(b), the bound of Property 1 is tight: $\epsilon = |f(x)|$. In particular $\delta = -f(x)\nabla_x f(x)$ is guaranteed to be an adversarial attack. The risk is the smallest possible. There is no classifier with the same risk and better certificates. Said otherwise the SDF(b) is the solution to:*

$$\max_{f \in Lip_1(\mathbb{R}^n, \mathbb{R})} \min_{x \in \mathcal{X}} \min_{\substack{\delta \in \mathbb{R}^n \\ sign(f(x+\delta)) \neq sign(f(x))}} \|\delta\|,$$
$$under\ the\ constraint\ f \in \underset{g \in Lip_1(\mathbb{R}^n, \mathbb{R})}{\arg\min} E(sign \circ g).$$

(3)

The SDF($b$) cannot be explicitly constructed since it relies on the (unknown) optimal Bayes classifier.

## 4.2 Improving the accuracy of the maximally robust classifier

On the opposite side, we exhibit a family of classifiers with lower accuracy but with higher certifiable robustness. We insist that the quantity of interest is the *certifiable robustness* $|f(x)|$ and not the *true empirical robustness* $\epsilon$ (which can be higher). The former is computed exactly and freely, while the latter is a difficult problem for which only upper bounds returned by attacks are available. In the literature, the robustness is only evaluated on well classified examples. The certificate can be both interpreted as a form of "confidence" of the network, and as the minimal perturbations required to switch the class. Hence, we shall weight negatively this certificate for the examples that are misclassified since confidence in presence of errors is worse. For this reason, we propose in Definition 5 a new metric called the Mean Certifiable Robustness (MCR).

**Definition 5** (**Mean Certifiable Robustness – MCR**)
*For any function $f : \mathcal{X} \to \mathbb{R} \in LipNet1$ we define its weighted mean certifiable robustness $\mathcal{R}_{(P,y)}(f)$*

*on class $P$ with label $y$ as:*

$$\mathcal{R}_{(P,y)}(f) := \mathbb{E}_{x \sim P}[\mathbb{1}\{yf(x) > 0\}|f(x)|] + \mathbb{E}_{x \sim P}[-\mathbb{1}\{yf(x) < 0\}|f(x)|] = \mathbb{E}_{x \sim P}yf(x).$$
(4)

We can readily see from the definition that the classifier with highest MCR for class $P$ is the constant classifier $f = y \times \infty$. The interest of this notion arises when we consider minimizing the loss function $\mathcal{L}^W(f(x), y) := -yf(x)$, i.e when looking for classifier with the highest MCR.

**Property 2. Wasserstein classifiers (i.e WGAN discriminators) are optimally robust**. *The minimum of $\mathcal{L}^W(f(x), y)$ over $P$ and $Q$ is the Wasserstein-1 distance [30] between $P$ and $Q$ according to the Kantorovich-Rubinstein duality:*

$$\max_{f \in Lip_1(\mathcal{X}, \mathbb{R})} \mathcal{R}_{(P,+1)}(f) + \mathcal{R}_{(Q,-1)}(f) = \min_{f \in Lip_1(\mathbb{R}^n, \mathbb{R})} \mathbb{E}_{\mathbb{P}_{XY}}[\mathcal{L}^W(f(x), y)] = \mathcal{W}_1(P, Q).$$
(5)

Even though the minimizer of $\mathcal{L}_W(f(x), y)$ can have low accuracy, it has the highest MCR. Interestingly, the minimizer $f^*$ of equation 5 is invariant by translation: $f^* - T$ is also a minimizer for any $T \in \mathbb{R}$. When $T \to \infty$ (resp. $-\infty$) the classifier has 100% recall on $Q$ (resp. $P$), and 0% on $P$ (resp. $Q$). Does it always exist $T^*$ with 100% accuracy overall? Sadly, even when the $P$ and $Q$ have disjoint support, the answer is no. We precise this empirical observation of [8] in Proposition 3.

**Proposition 3. WGAN discriminators are weak classifiers**. *For every $\frac{1}{2} \geq \epsilon > 0$ there exist distributions $P$ and $Q$ with disjoint supports in $\mathbb{R}$ such that for any optimum $f$ of equation 5, the error of classifier $sign \circ f$ is superior to $\frac{1}{2} - \epsilon$.*

Note that this minimum also invariant by dilatation: any *finite* upper bound $L$ can be chosen for Equation 5 (see Appendix G).

## 4.3 Controlling the accuracy/robustness tradeoff with loss parameters

Now that the extrema of the accuracy robustness tradeoff were characterized in 4.1 and 4.2, is yet to be answered if it is possible to control this tradeoff using conventional loss (and its parameters, as introduced in 3.2).

Interestingly, observe that $\mathcal{L}_\tau^{bce}(f(x), y) = \log 2 - \frac{y\tau f(x)}{2} + \mathcal{O}(\tau^2 f^2(x))$ so when $\tau \to 0$ we get:

$$\min_{f \in \text{Lip}_1(\mathcal{X}, \mathbb{R})} \frac{4}{\tau} \left( \mathbb{E}_{(x,y) \sim P_{XY}}[\mathcal{L}_\tau^{bce}(f(x), y)] - \log 2 \right) = -\mathcal{W}_1(P, Q).$$

In the limit of small temperatures, the BCE minimizer essentially behaves like the classifier of the highest MCR (see Figure 3 and Appendix H). Similarly, the HKR loss $\mathcal{L}^{hkr}$ introduced in [8] for LipNet1 training allows fine grained control of the accuracy-robustness tradeoff:

$$\mathcal{L}_{m,\alpha}^{hkr}(f(x), y) = \mathcal{L}^W(f(x), y) + \alpha \mathcal{L}_m^H(f(x), y) = -yf(x) + \alpha \max(0, m - yf(x)). \quad (6)$$

We recover $\mathcal{W}_1$ behavior for $\alpha = 0$, and hinge $\mathcal{L}_m^H$ behavior for $\alpha \to \infty$, in a fashion that reminds the role of $\tau$ for $\mathcal{L}^{bce}$.

A key takeaway is that BCE, HKR and hinge loss have parameters that allow to control the accuracy robustness tradeoff, reaching on one side the maximum robustness of MCR, and the accuracy of unconstrained networks on the other. Empirically this tradeoff is observed as a Pareto front with accuracy on one axis, and robustness on the other. Figure 3 shows this on the CIFAR10 dataset using the $\epsilon$ robustness as robustness measures (other robustness measure yield similar observations, see fig 10a and 10b).

In conclusion, these last two sections demonstrate that restraining networks to be in LipNet1 does not impact the classification capabilities while providing certificates of robustness; however, for these networks the loss parameters play an important role in this trade-off.

## 5 1-Lipschitz classifiers have generalization guarantees

In this section, we explore the statistical and optimization properties of LipNet1 networks, and we prove the assumption of [31] that "adjusting the Lipschitz constant of a feed-forward neural network controls how well the model will generalise to new data".

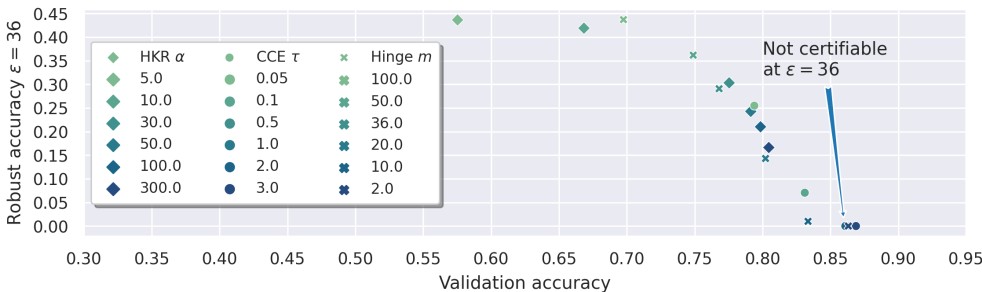

Figure 3: **Accuracy-Robustness trade-off on CIFAR10 with Hinge, HKR and Categorical Cross-Entropy (CCE) hyper-parameters.** Overall, for a given network architecture, a Pareto front appears between clean accuracy and robust accuracy. We move along it by tuning the parameters of each loss. We trained small LipNet1 *CNNs* (0.4M params) with basic data augmentation (see appendix K for detailed experimental setting).

## 5.1 Consistency of LipNet1 class

LipNet1 class enjoys another remarkable property since it is a Glivenko-Cantelli class: minimizers of Lipschitz losses are consistent estimators. In other words, as the size of the training set increases, the training loss becomes a proxy for the test loss: LipNet1 neural networks will not overfit in the limit of (very) large sample sizes.

**Proposition 4. Train Loss is a proxy of Test Loss**. *Let $\mathbb{P}_{XY}$ a probability measure on $\mathcal{X} \times \mathcal{Y}$ where $\mathcal{X} \subset \mathbb{R}^n$ is a bounded set. Let $(x_i, y_i)_{1 \leq i \leq p}$ be a sample of $p$ iid random variables with law $\mathbb{P}_{XY}$. Let $\mathcal{L}$ be a Lipschitz loss function over $\mathbb{R} \times \mathcal{Y}$. We define:*

$$\mathcal{E}_p(f) := \frac{1}{p} \sum_{i=1}^{p} \mathcal{L}(f(x_i), y_i) \text{ and } \mathcal{E}_\infty(f) := \mathbb{E}_{(x,y) \sim \mathbb{P}_{XY}}[\mathcal{L}(f(x), y)]. \tag{7}$$

*Then the empirical loss $\mathcal{E}_p(f)$ converges to the test loss $\mathcal{E}_\infty(f)$ (taking the limit $p \to \infty$):*

$$\min_{f \in Lip_L(\mathcal{X}, \mathbb{R})} \mathcal{E}_p(f) \xrightarrow{a.s} \min_{f \in Lip_L(\mathcal{X}, \mathbb{R})} \mathcal{E}_\infty(f). \tag{8}$$

It is another flavor of the bias-variance trade-off in learning. Thanks to Corollary 1 we know the LipNet1 class does not suffer of bias, while the generalization gap (i.e the variance) can be made as small as we want by increasing the size of the training set (see Figure 4). The number of examples required to close the generalization gap is dataset specific in general, however it seems that with low $\tau$ fewer examples are required. This result may seem obvious, but we emphasize **this property is not shared by AllNet networks** (see Proposition 9 in Appendix C.2). Nonetheless, most practitioners take for granted that bigger training sets ensure generalization for AllNet networks.

## 5.2 Understanding why unconstrained networks are prone to overfitting

Surprisingly, on AllNet networks, minimization of BCE leads to uncontrolled growth of Lipschitz constant and saturation of the predicted probabilities. This is an impediment to generalization results.

**Proposition 5. Optimizing BCE over AllNet leads to divergence**. *Let $f_t$ be a sequence of neural networks, that minimizes the BCE over a non-trivial training set (at least two different examples with different labels) of size $p$, i.e assume that:*

$$\lim_{t \to \infty} \frac{1}{p} \sum_{i=1}^{p} \mathcal{L}_\tau(f_t(x_i), y_i) = 0. \tag{9}$$

*Let $L_t$ be the Lipschitz constant of $f_t$. Then $\lim_{t \to \infty} L_t = +\infty$. There is at least one weight matrix $W$ such that $\lim_{t \to \infty} \|W_t\| = +\infty$. Furthermore, the predicted probabilities are saturated:*

$$\lim_{t \to \infty} \sigma(f_t(x_i)) \in \{0, 1\}. \tag{10}$$

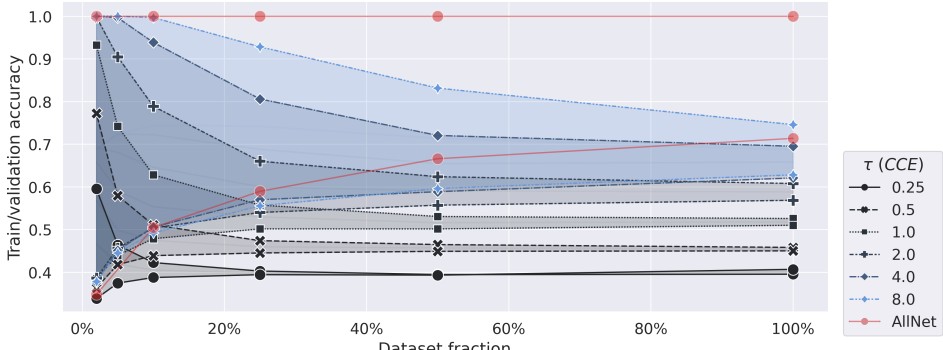

Figure 4: **Link between LipNet1 and generalization gap, dataset size and cross-entropy temperature.** We train a CNN on different fractions of the CIFAR10 train set (2%, 5%, 10%, 25%, 50% and 100% on $x$-axis) with different values of temperature $\tau$ (highlighted by different colors). Train (resp. validation) accuracy forms the upper (resp. lower) bound of each envelope. As $\tau$ increases, more samples are required to reduce the generalization gap. Conversely, training a LipNet1 network with small $\tau$ is equivalent to training a Lipschitz network with small $L$: the network generalizes well but the accuracy reaches a plateau (under-fitting). The AllNet network (in red) severely overfit: the generalization gap is large and validation accuracy corresponds to the limit that would reach a LipNet1 as $\tau$ increases. See appendix J for detailed experimental setting.

This issue is especially important since Lipschitz constant and adversarial vulnerabilities are related [32]. The predicted probability $\sigma(f(x))$ will either be 0 or 1 (regardless of the train set), which do not carry any useful information on the true confidence of the classifier

**Example 1.** *Consider a classification task on $\mathbb{R}$ with linearly separable inputs $\{-1, 1\}$ and labels $\{-1, 1\}$. We use an affine model $f(x) = Wx + b$ for the logits (with $W \in \mathbb{R}$ and $b \in \mathbb{R}$) (one-layer neural network). It exists $\bar{W}, \bar{b}$ such that $f$ achieves $100\%$ accuracy. However, as noticed in [33] (Section 4.3.2) the BCE loss will not be zero. The minimization occurs only with the diverging sequence of parameters $(\lambda \bar{W}, \lambda \bar{b})$ as $\lambda \to \infty$. It turns out the infimum is not a minimum!*

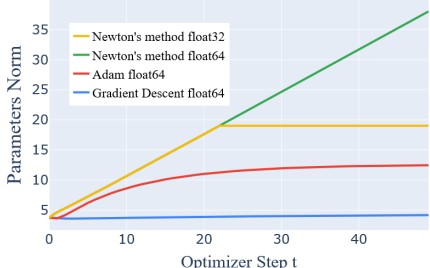

Figure 5

Even on toy example 1 with a trivial model, the minimization problem is ill-defined. Without weight regularization, the minimizer can not be attained. This is compliant with the high Lipschitz constant of AllNet networks that have been observed in practice [23], and is confirmed by our experiment on MNIST with a ConvNet (see Figure 12). The spectral norm of the weights is multiplied by 5 over the course of 25 epochs, whereas the validation accuracy remains the same (around 99%).

Furthermore, there is an issue of vanishing gradients with BCE : first order methods struggle to saturate the logits of AllNet networks, whereas second order methods in *float64* diverge as expected. The poor properties of the optimizer, and the rounding errors in 32 bits floating point arithmetic, have greatly contributed to the caveat of BCE minimization remaining mostly unnoticed by the community.

## 5.3 Lipschitz classifiers are PAC learnable

Hinge loss $\mathcal{L}_m^H$ and HKR loss $\mathcal{L}^{hkr}$ benefit from Proposition 4. The certificate $|f(x)|$ can be understood as confidence. Hence, we are interested in a classifier that makes a decision only if the prediction is above some threshold $m > 0$, while $|f(x)| < m$ can be understood as examples $x$ for which the classifier is unsure: the label may be flipped using attacks of norm $\epsilon \leq m$. In this setting, we fall back to PAC learnability [10]: this theory gives bounds on the number of train samples

| Properties | | AllNet network | LipNet1 network |
|---|---|---|---|
| Fit any boundary | | yes [13] | yes (Proposition 1) |
| Robustness certificates | | no | yes (Property 1) |
| Consistent estimator | | no (App C.2) | yes (Proposition 4, Figures 1b, 4) |
| Gradients | | exploding or vanishing | preserved for GNP (App F) |
| VC dimension bounds | | architecture dependent [35] | when $m > 0$ (Proposition 6) |
| BCE $\mathcal{L}_\tau^{bce}$ | minimizer | ill-defined $L_t \to \infty$ (Proposition 5) | attained (Proposition 2) |
| | remark | vanishing gradient (Ex 1) | $L$ or $\tau$ must be tuned (Figure 3) |
| Wasserstein $\mathcal{L}^W$ | minimizer | ill-defined $L_t \to \infty$ | attained, robust (Property 2) |
| | remark | diverges during training | weak classifier (Proposition 3) |
| Hinge $\mathcal{L}_m^H$ | minimizer | attained | attained |
| | remark | no guarantees on margin | $m$ must be tuned |
| HKR $\mathcal{L}_{m,\alpha}^{hkr}$ | minimizer | ill-defined $L_t \to \infty$ | accuracy-robustness tradeoff |
| | remark | diverges during training | $\alpha$ and $m$ must be tuned (Figure 3) |

Table 1: Summary of notable results and the contributions.

required to guarantee that the test error will fall below some threshold $0 \leq e < \frac{1}{2}$ with probability at least $1 > \beta \geq 0$, through the use of Vapnik Chervonenkis (VC) dimension bounds [34].

**Proposition 6. 1-Lipschitz Functions with margin are PAC learnable**. *Assume $P$ and $Q$ have bounded support $\mathcal{X}$. Let $m > 0$ the margin. Let $\mathcal{C}^m(\mathcal{X}) = \{c_f^m : \mathcal{X} \to \{-1, \perp, +1\}, f \in Lip_1(\mathcal{X}, \mathbb{R})\}$ be the hypothesis class defined as follow.*

$$c_f^m(x) = \begin{cases} +1 & \text{if } f(x) \geq m, \\ -1 & \text{if } f(x) \leq -m, \\ \perp & \text{otherwise, meaning "f doesn't feel confident".} \end{cases} \quad (11)$$

*Let $\mathfrak{B}$ be the unit ball. Then the VC dimension of $\mathcal{C}^m$ is finite:*

$$(\frac{1}{m})^n \frac{vol(\mathcal{X})}{vol(\mathfrak{B})} \leq VC_{dim}(\mathcal{C}^m(\mathcal{X})) \leq (\frac{3}{m})^n \frac{vol(\mathcal{X})}{vol(\mathfrak{B})}. \quad (12)$$

Interestingly if the classes are $\epsilon$ separable ($\epsilon > 0$), choosing $m = \epsilon$ guarantees that 100% accuracy is reachable. Prior over the separability of the input space is turned into VC bounds over the space of hypothesis. When $m = 0$ the VC dimension of space $\mathcal{C}^m(\mathcal{X})$ becomes infinite and the class is not PAC learnable anymore: the training error will not converge to test error in general, regardless of the size of the training set. It is not a contradiction with Proposition 4: error $E(c_f^m(x))$ lacks continuity w.r.t $f(x)$ so it is not a consistent estimator.

This VC bound is *architecture independent* which contrasts with the rest of literature on AllNet networks. Practically, it means that the LipNet1 network architecture can be chosen as big as we want without risking overfitting, as long as the margin $m$ is chosen appropriately. Proposition 7 also provides an architecture dependant bound for LipNet1 networks.

**Proposition 7. VC dimension of LipNet1 neural networks**. *Let $f_\theta : \mathbb{R}^n \to \mathbb{R}$ a LipNet1 neural network with parameters $\theta \in \Theta$, with **GroupSort2** activation functions, and a total of $W$ neurons. Let $\mathcal{H} = \{sign f_\theta | \theta \in \Theta\}$ the hypothesis class spanned by this architecture. Then we have:*

$$VC_{dim}(\mathcal{H}) = \mathcal{O}\left((n+1)2^W\right). \quad (13)$$

In literature, tighter VC dimension bounds for neural networks exist, but they assume element-wise activation function [35]. This hypothesis does not apply to GroupSort2 which is known to be more expressive [36], however we believe that this preliminary result can be strengthened.

## 6 Related work

**LipNet1 networks parametrization** benefit from a rich literature (see Appendix D) to enforce the Lipschitz constraint in various layers [37, 38, 6, 39, 40, 14, 41, 42, 43, 44] such as activation

functions, affine layers, attention layers or recurrent units. Residual connections are also Lipschitz (see Appendix F). **Gradient Norm Preserving networks** avoid the vanishing gradients [3, 45] phenomenon to which the LipNet1 networks are prone, by using orthogonal matrices in affine layers. This justifies the *"orthogonal neural network"* terminology [46, 47, 48]. ReLU based Lipschitz networks suffer from expressiveness issues [11], and activation functions like GroupSort [11, 36] (a special case of Householder reflection [49, 50]) have been proposed in replacement. **Orthogonal kernels** are still an active research area [51, 52, 53, 54, 3, 55, 56, 57]. They are used in normalizing flows [58], ensemble methods [59], reinforcement learning [60] or graph neural networks [61]. The optimization over the group of orthogonal matrices (known as Stiefel manifold) has been extensively studied in [62], and algorithms suitable for deep learning are detailed in [63, 64, 64, 65, 66, 67, 68, 69].

**Generalization bounds** for general Lipschitz classifiers are given in [70, 71, 72]. Links between adversarial robustness, large margins classifiers and optimization bias are studied in [73, 74, 16, 75]. The importance of the loss in adversarial robustness is studied in [76]. See Appendix C.5.

## 7    Conclusions

In this paper, we challenged the common belief that constraining Lipschitz constant degrades the classification performance of neural networks. We proved that LipNet1 networks exhibit numerous attractive properties (see Table 1 in summary): they provide robustness radius certificates without restrictions on their expressive power. They benefit from generalization guarantees. We showed that the hidden parameters of the loss allow to control the generalization gap and certifiable robustness.

While the question of the LipNet1 architecture is often in the spotlight, the loss is overlooked. We pointed out that Cross-Entropy is not necessarily the best choice, margin-based losses, such as hinge or its variant HKR, have appealing properties (table 1).

## 8    Perspectives

This paper aims to be at the intersection between theoretical ML and (empirical) deep learning. Lipschitz constrained networks allow to directly put in perspective mathematical proofs and we are confident that this theory can be verified empirically on very large-scale vision datasets (such as Imagenet [77]).

This paper also provides a toolbox of results and experiments to serve as a basis for future works. We aim to open new research directions, including outside the field of robust learning. AllNet networks could benefit from LipNet1 literature: the absence of control over the Lipschitz constant of AllNet is mitigated in practice by elements such as mixup or weight decay. Such elements would be better understood by looking at how they affect the (uncontrolled) Lipschitz constant of AllNet .

The efficient training over LipNet1 is still an active research area. Moreover, AllNet networks benefits from architectural elements such as skip connections and batch normalization (see appendix F). As LipNet1 networks get more mature, empirical results will improve, matching theory even more (explaining the emphasis on the theoretical proofs instead of the design of LipNet1 depicted in appendix D).

Many practices in deep learning entangle the questions of architecture, of generalization and of optimization. However, these elements usually have unexpected consequences on the nature of the optimum and the optimization process. Our work is a first step toward a better separation of these components and their role.

## Acknowledgments and Disclosure of Funding

We thank Sébastien Gerchinovitz for critical proof checking, Jean-Michel Loubes for useful discussions, and Etienne de Montbrun, Thomas Fel and Antonin Poché for their read-checking. A special thank to Agustin Picard for his useful advice and thorough reading of the paper. This work has benefited from the AI Interdisciplinary Institute ANITI, which is funded by the French "Investing for the Future – PIA3" program under the Grant agreement ANR-19-P3IA-0004. The authors gratefully acknowledge the support of the DEEL project.[2]

---

[2] https://www.deel.ai/

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
