# Contents

## A  Proofs of Section 3

### A.1  Single class case

The properties of Lebesgue measure over euclidean space $\mathbb{R}^n$ are recalled in [78]. Importantly, the Lebesgue measure $\mu$ is translation invariant and measure the volume of hyperboxes: $\mu([a_1, b_1], [a_2, b_2], \dots [a_n, b_n]) = \Pi_{i=1}^n (b_i - a_i)$.

**Proposition 1. Lipschitz Binary classification.** *For any binary classifier $c : \mathcal{X} \to \mathcal{Y}$ with closed pre-images ($c^{-1}(\{y\})$ is a closed set) there exists a 1-Lipschitz function $f : \mathbb{R}^n \to \mathbb{R}$ such that $\text{sign}(f(x)) = c(x)$ on $\mathcal{X}$ and such that $\|\nabla_x f\| = 1$ almost everywhere (w.r.t Lebesgue measure).*

The proof of **Proposition 1** is constructive, we need to introduce the Signed Distance Function, already popularized in shape processing [25].

**Definition 6**
**Signed Distance Function associated to decision boundary.** *Let $c : \mathcal{X} \to \{-1, +1\}$ be any classifier with closed pre-images. Let $\bar{A} = \{x \in \mathbb{R}^n | c(x) = +1\}$ and $\bar{B} = \{x \in \mathbb{R}^n | c(x) = -1\} = \mathcal{X} \setminus \bar{A}$. Let $d(x, y) = \|x - y\|$ and $d(x, S) = \min_{y \in S} d(x, y)$ be the distance to a **closed set** $S$. Let $\partial = \{x \in \mathbb{R}^n | d(x, \bar{A}) = d(x, \bar{B})\}$. We define $f : \mathbb{R}^n \to \mathbb{R}$ as follow:*

$$f(x) = \begin{cases} d(x, \partial) & \text{if } d(x, \bar{B}) \geq d(x, \bar{A}) \\ -d(x, \partial) & \text{if } d(x, \bar{B}) < d(x, \bar{A}). \end{cases} \tag{14}$$

*We denote by SDF(c) the function $f$.*

The signed distance function $f$ previously defined verifies all the properties, as a special case of Eikonal equation. We give the full proof here for completeness.

*Proof.* We start by proving that $f$ is 1-Lipschitz. The reasoning applies more broadly to arbitrary Banach space (topological normed vector space), not only $(\mathbb{R}^n, \| \cdot \|_2)$.

First, consider the case $d(x, \bar{B}) \geq d(x, \bar{A})$ and $d(y, \bar{B}) \geq d(y, \bar{A})$. Then we have $|f(x) - f(y)| = |d(x, \partial) - d(y, \partial)|$. Assume without loss of generality that $d(x, \partial) \geq d(y, \partial)$. Let $z \in \partial$ be such that $d(y, \partial) = d(y, z)$ (it is guaranteed to exist since $\partial$ is a closed set). Then by definition of $d(x, \partial)$ we have $d(x, z) \geq d(x, \partial)$. So:

$$|f(x) - f(y)| = |d(x, \partial) - d(y, \partial)| \leq d(x, z) - d(y, z) \leq d(x, y). \tag{15}$$

The cases $d(x, \bar{B}) < d(x, \bar{A})$ and $d(y, \bar{B}) < d(y, \bar{A})$ are identical. Now consider the case $d(x, \bar{B}) < d(x, \bar{A})$ and $d(y, \bar{B}) \geq d(y, \bar{A})$. Then we have $|f(x) - f(y)| = d(x, \partial) + d(y, \partial)$. We will proceed

by contradiction. Such complicated reasoning is superfluous for $(\mathbb{R}^n, \|\cdot\|_2)$, but has the appealing property to generalize to any Banach space. Assume that $d(x, \partial) + d(y, \partial) > d(x, y)$. Let $R > 0$ be such that $R < d(x, \partial)$ and $R + d(y, \partial) > d(x, y)$. Let:

$$z = x + \frac{R}{d(x, y)}(x - y).$$

Then we have $d(x, z) = \|\frac{R}{d(x,y)}(x - y)\| = \frac{R}{d(x,y)}d(x, y) = R < d(x, \partial)$. So by definition of $\partial$ we have $d(z, \bar{B}) < d(z, \bar{A})$. But we also have:

$$d(y, z) = \|(x - y) + \frac{R}{d(x, y)}(x - y)\| = |1 - \frac{R}{d(x, y)}| \times \|x - y\|.$$
$$= |d(x, y) - R| < |d(y, \partial)| \text{ using the hypothesis on } R. \tag{16}$$

So we have $d(z, \bar{B}) \geq d(z, \bar{A})$ which is a contradiction. Consequently, we must have $d(x, \partial) + d(y, \partial) \leq d(x, y)$. The function $f$ is indeed 1-Lipschitz.

Now, we will prove that $\|\nabla_x f\| = 1$ everywhere it is defined. Let $x$ be such that $y \in \arg\min_{y \in \partial} d(x, y)$ is unique. Consider $h = \epsilon \frac{(y - x)}{\|y - x\|}$ with $1 \geq \epsilon > 0$ a small positive real. We have $d(x, x + h) = \epsilon$, it follows by triangular inequality that $d(x + h, \partial) = d(x, \partial) - \epsilon$. We see that:

$$\lim_{\epsilon \to +\infty} \frac{f(x + h) - f(x)}{\|h\|} = -1.$$

The vector $u = -\nabla_x f$ is the (unique) vector for which $\langle u, \frac{f(x+h) - f(x)}{\|h\|} \rangle$ is minimal. Knowing that $f$ is 1-Lipschitz yields that $\|\nabla_x f\| = 1$. For points $x$ for which $\arg\min_{y \in \partial} d(x, y)$ is not unique, the gradient is not defined because different directions minimize $\langle u, \frac{f(x+h) - f(x)}{\|h\|} \rangle$ which contradicts the uniqueness of gradient vector. The number of points for which $y \in \arg\min_{y \in \partial} d(x, y)$ is not unique must have null measure, since Lipschitz functions are almost everywhere differentiable (by Rademacher's Theorem).

Finally, note that $\text{sign} f(x) = c(x)$ on $\bar{A}$ and $\bar{B}$. Indeed, in this case either $d(x, \bar{B}) < d(x, \bar{A})$ either $d(x, \bar{B}) > d(x, \bar{A})$ and the result is straightforward. $\square$

## A.2 Multiclass case

The label set is now $\mathcal{Y} = \{1, 2, \ldots, K\}$. In practice we use one-hot encoded vectors to compute the loss, by taking the $\arg\max_k$ over a vector of $\mathbb{R}^K$.

**Proposition 8** (Lipschitz Multiclass classification). *For any multiclass classifier $c : \mathcal{X} \to \mathcal{Y}$ with closed pre-images there exists a 1-Lipschitz function $f : \mathbb{R}^n \to \mathbb{R}^K$ such that $\arg\max_k f_k(x) = c(x)$ on $\mathcal{X}$ and such that $\|J_x f\| = 1$ almost everywhere (w.r.t Lebesgue measure).*

For the case $K > 2$ we must slightly change the definition to prove **Proposition 8**.

**Definition 7** (Multiclass Signed Distance Function)
*Let $c : \mathcal{X} \to \{1, 2, \ldots K\}$ be any classifier with closed pre-images. Let $\bar{A}_k = c^{-1}(\{k\})$. Let $\partial = \{x \in \mathbb{R}^n | \exists k \neq l, d(x, \bar{A}_k) = d(x, \bar{A}_l) = \arg\min_m d(x, \bar{A}_m)\}$. We define $f : \mathbb{R}^n \to \mathbb{R}^k$ as follow:*

$$f_k(x) = \begin{cases} d(x, \partial) & \text{if } d(x, \bar{A}_k) < d(x, \bar{A}_l) \text{ for all } l \neq k, \\ 0 & \text{otherwise.} \end{cases} \tag{17}$$

In overall the proof remains the same.

*Proof.* We start by proving that $f$ is 1-Lipschitz.

We need to prove that $\|f(x) - f(y)\|_p \leq \|x - y\|$ for any $p$-norm on $\mathbb{R}^K$ with $p \geq 1$. First, consider the case $f_k(x) = f_k(y) \neq 0$. Then $\|f(x) - f(y)\|_p = |f_k(x) - f_k(y)| = |d(x, \partial) - d(y, \partial)| \leq \|x - y\|$ using the proof of Proposition 1. Now, consider the case $f_k(x) > 0$, $f_l(y) > 0$ and $k \neq l$. Then:

$$\|f(x) - f(y)\|_p = \sqrt[p]{f_k^p(x) + f_l^p(y)} \leq |f^k(x)| + |f^l(y)| = d(x, \partial) + d(y, \partial). \tag{18}$$

Using the same technique as in the previous proof, if we assume $d(x, \partial) + d(y, \partial) > d(x, y)$ then we can construct $z$ verifying both $f_k(z) < f_l(z)$ and $f_l(z) < f_k(z)$ which is a contradiction. Consequently $d(x, \partial) + d(y, \partial) \leq d(x, y)$.

Each row of $J_x f$ is either full of zeros, or the gradient of some $f_k$ on which the reasoning of the case $K = 2$ applies (like in the previous proof). In this case, the spectral norm $J_x f$ is equal to the norm of the gradient of the non zero row. We conclude similarly that $\|J_x f\| = 1$ everywhere it is defined.

Finally, note that $\arg\max_k f_k(x)$ is equal to $c(x)$ everywhere $c$ is defined, which concludes the proof. $\qquad\square$

With this proposition in mind, we can deduce Corollary 3.

**Corollary 3** (LipNet1 is as powerful as AllNet for classification). *For any neural network $f : \mathbb{R}^n \to \mathbb{R}$ there exists 1-Lipschitz neural network $\tilde{f} : \mathbb{R}^n \to \mathbb{R}$ such that $\text{sign}(f(x)) = \text{sign}(\tilde{f}(x))$.*

*Proof.* The proof sketched in Introduction is sufficient to show that LipNet1 networks and unconstrained ones have the same decision frontiers. We could have also taken a more convoluted path: take the classifier $c$ associated to an AllNet network, consider the restriction to a subset $\mathcal{X}$ of the input space making the pre-images separated. Then we can apply Proposition 1 to get a 1-Lipschitz function with the same classification power, and finally approximate those functions (in the sense of uniform convergence) with LipNet1 network. $\qquad\square$

**Corollary 1. Separable classes implies zero error**. *If $P$ and $Q$ are $\epsilon$-separated, then there exists a network $f \in$ LipNet1 such that **error** $E(\text{sign} \circ f) := \mathbb{E}_{(x,y) \sim \mathbb{P}_{XY}}[\mathbb{1}\{\text{sign}(f(x)) \neq y\}] = 0$.*

*Proof.* If classes are separable the optimal Bayes classifier $b$ achieves zero error. Moreover, the topological closure $\overline{b^{-1}}(\{y\}), y \in \mathcal{Y}$ yields a set of closed sets that are all disjoints (since $\epsilon > 0$) and on which Proposition 8 can be applied, yielding a LipNet1 neural network with the wanted properties.

**Bonus: non separable case.** We can also handle the case of non separable classes by imitating the optimal Baye classifier $c$. We take $\mathcal{X}$ a subset of the input space on which the pre-images of $c$ are closed. The application of Proposition 1 for optimal Bayes classifier gives us a 1-Lipschitz function $f$ with same decision frontier as $c$. Finally, we can use the universal approximation theorem of Anil and all. in [11] to conclude there exists LipNet1 network that can approximate arbitrary well the function $f$, and hence approximate arbitrarily well the classifier $c$ on $\mathcal{X}$. Outside $\mathcal{X}$, the error is not controlled but depends of the volume of the set $(\text{supp } \mathbb{P}_X)/\mathcal{X}$ whose Lebesgue measure can be made arbitrary small (by taking $\mathcal{X}$ big enough). As $\mathbb{P}_X$ admits a pdf w.r.t Lebesgue measure, then $\mathbb{P}_X((\text{supp } \mathbb{P}_X)/\mathcal{X})$ can be made arbitrary small, and consequently the risk as well. $\qquad\square$

**Example 1a.** We plot the level set of the network $f$ trained from the discretized ground truth (in $400 \times 400$ pixels) of the Signed distance function. The distance to the frontier $\partial$ is easily computed since the frontier $\partial$ is a finite collection of segments (fourth iteration of Von Koch snowflake fractal). We train a $128 \to 128 \to 128 \to 128 \to 128$ LipNet1 network. The network is trained with Mean Square Error (MSE) and the criterion used to stop training is the Mean Absolute Error (MAE).

### A.3 Proofs of Section 3.2

**Proposition 2. BCE minimization for 1-Lipschitz functions**. *Let $\mathcal{X} \subset \mathbb{R}^n$ be a compact and $\tau > 0$. Then the infimum in Equation 2 is a minimum, denoted $f^\tau \in \text{Lip}_1(\mathcal{X}, \mathbb{R})$:*

$$f^\tau \in \arg\inf_{f \in \text{Lip}_1(\mathcal{X}, \mathbb{R})} \mathbb{E}_{(x,y) \sim \mathbb{P}_{XY}}[\mathcal{L}^{bce}_\tau(f(x), y)]. \tag{2}$$

*Moreover, the LipNet1 networks will not suffer of vanishing gradient of the loss (see Appendix F).*

The proof of **Proposition 2** is an application of Arzelà–Ascoli theorem.

*Proof.* Let $\mathcal{E}(f) = \mathbb{E}_{(x,y) \sim \mathbb{P}_{XY}}[\mathcal{L}(f(x), y)]$. Consider a sequence of functions $f^t$ in $\text{Lip}_L(\mathcal{X}, \mathbb{R})$ such that $\lim_{t \to \infty} \mathcal{E}(f_t) = \inf_{f \in \text{Lip}_L(\mathcal{X}, \mathbb{R})} \mathcal{E}(f) = \mathcal{E}^*$.

Consider the sequence $u_t = \|f_t\|_\infty$. We want to prove that $(u_t)_{t \in \mathbb{N}}$ is bounded. Proceed by contradiction and observe that if $\limsup_{t \to \infty} u_t = +\infty$ then $\limsup_{t \to \infty} \mathcal{E}(f_t) = +\infty$. Indeed, for $\|f_t\|_\infty \geq 2L\mathrm{diam}\,\mathcal{X}$ we can guarantee that $\mathrm{sign}\,f_t$ is constant over $\mathcal{X}$ and in this case one of the two classes $y$ is misclassified, knowing that $\lim_{f(x) \to \infty} \mathcal{L}(-yf(x), y) = \mathcal{O}(f(x)) \to +\infty$ yields the desired result.

But if $\limsup_{t \to \infty} \mathcal{E}(f_t) = +\infty$, then $\mathcal{E}(f_t)$ cannot not converges to $\mathcal{E}^*$. Consequently, $u_t$ must be upper bounded by some $M$.

Hence the sequence $f_t$ is uniformly bounded. Moreover each function $f_t$ is $L$-Lipschitz so the sequence $f_t$ is uniformly equicontinuous. By applying Arzelà–Ascoli theorem we deduce that it exists a subsequence $f_{\phi(t)}$ (where $\phi : \mathbb{N} \to \mathbb{N}$ is strictly increasing) that converges uniformly to some $f^*$, and $f^* \in \mathrm{Lip}_L(\mathcal{X}, \mathbb{R})$. As $\mathcal{E}(f^*) = \mathcal{E}^*$, the infimum is indeed a minimum. $\qquad\square$

The upper bound on $\mathrm{Lip}(f)$ is turned into a lower bound on $\|\nabla_\theta \mathcal{L}(f_L^{\theta^*}(x), y)\|$ (no element-wise vanishing gradient), but its average $\|\nabla_\theta \mathbb{E}_{(x,y) \sim \mathbb{P}_{XY}}[\mathcal{L}((f_L^{\theta^*}(x), y)]\| = 0$ is null (see Appendix F).

# B   Proofs of Section 4

We recall below the definition of the Signed Distance Function (SDF) associated to a classifier.

**Definition 6**

**Signed Distance Function associated to decision boundary.** *Let $c : \mathcal{X} \to \{-1, +1\}$ be any classifier with closed pre-images. Let $\bar{A} = \{x \in \mathbb{R}^n | c(x) = +1\}$ and $\bar{B} = \{x \in \mathbb{R}^n | c(x) = -1\} = \mathcal{X} \setminus \bar{A}$. Let $d(x, y) = \|x - y\|$ and $d(x, S) = \min_{y \in S} d(x, y)$ be the distance to a **closed** set $S$. Let $\partial = \{x \in \mathbb{R}^n | d(x, \bar{A}) = d(x, \bar{B})\}$. We define $f : \mathbb{R}^n \to \mathbb{R}$ as follow:*

$$f(x) = \begin{cases} d(x, \partial) & \text{if } d(x, \bar{B}) \geq d(x, \bar{A}) \\ -d(x, \partial) & \text{if } d(x, \bar{B}) < d(x, \bar{A}). \end{cases} \tag{14}$$

*We denote by SDF(c) the function $f$.*

In proof of Corollary 2 we use the Bayes classifier $b : \mathcal{X} \to \mathcal{Y}$ associated to the classification task between $P$ and $Q$.

**Corollary 2.** *For the SDF(b), the bound of Property 1 is tight: $\epsilon = |f(x)|$. In particular $\delta = -f(x)\nabla_x f(x)$ is guaranteed to be an adversarial attack. The risk is the smallest possible. There is no classifier with the same risk and better certificates. Said otherwise the SDF(b) is the solution to:*

$$\max_{f \in \mathrm{Lip}_1(\mathbb{R}^n, \mathbb{R})} \min_{x \in \mathcal{X}} \min_{\substack{\delta \in \mathbb{R}^n \\ sign(f(x+\delta)) \neq sign(f(x))}} \|\delta\|,$$

$$\text{under the constraint } f \in \arg\min_{g \in \mathrm{Lip}_1(\mathbb{R}^n, \mathbb{R})} E(sign \circ g). \tag{3}$$

*Proof.* Those properties hold by construction. The risk $\mathcal{R}(\mathrm{sign}(f))$ is minimal since $f$ is build with the optimal Bayes classifier. Note that, in general, for any classifier $c : \mathcal{X} \to \mathcal{Y}$ the bound of Property 1 is tight by construction for SDF($c$). Indeed $f(x)$ is the distance to the frontier, and the direction is given by $\nabla_x f(x)$. $\qquad\square$

For the proof of Proposition 2 we recall below the definition of Wasserstein-1 distance, as found in [30] (Definition 6.1).

**Definition 8** (Wasserstein-1 distance)

*Let $d : \mathbb{R}^n \times \mathbb{R}^n \to \mathbb{R}$ be a metric. For any two measures $P$ and $Q$ on $\mathbb{R}^n$ the Wasserstein-1 distance is defined by the following formula:*

$$\mathcal{W}_1(P, Q) := \inf_{\pi \in \Pi(P,Q)} \int_{\mathbb{R}^n} d(x, y) \mathrm{d}\pi(x, y)$$

*where $\Pi(P, Q)$ denote the set of measures on $\mathbb{R}^n \times \mathbb{R}^n$ whose marginals are $P$ and $Q$ respectively. Equivalently we can write:*

$$\mathcal{W}_1(P, Q) := \inf_{\substack{Law(X)=P \\ Law(Y)=Q}} \mathbb{E}[d(X, Y)].$$

In our case we are working with neural networks that are Lipschitz with respect to $l2$ distance, so we have $d(x, y) := \|x - y\|_2$.

**Property 2. Wasserstein classifiers (i.e WGAN discriminators) are optimally robust**. *The minimum of $\mathcal{L}^W(f(x), y)$ over $P$ and $Q$ is the Wasserstein-1 distance [30] between $P$ and $Q$ according to the Kantorovich-Rubinstein duality:*

$$\max_{f \in Lip_1(\mathcal{X}, \mathbb{R})} \mathcal{R}_{(P, +1)}(f) + \mathcal{R}_{(Q, -1)}(f) = \min_{f \in Lip_1(\mathbb{R}^n, \mathbb{R})} \mathbb{E}_{\mathbb{P}_{XY}}[\mathcal{L}^W(f(x), y)] = \mathcal{W}_1(P, Q). \quad (5)$$

*Proof.* The result is straightforward by writing the dual formulation (following Kantorovich-Rubinstein) of Wasserstein $\mathcal{W}_1$ metric.

By Remark 6.3 of [30] the Wasserstein-1 distance is the Kantorovich-Rubinstein distance:

$$\mathcal{W}_1(P, Q) = \sup_{f \in \mathrm{Lip}_1(\mathcal{X}, \mathbb{R})} \mathbb{E}_{x \sim P}[f(x)] + \mathbb{E}_{z \sim Q}[f(z)]$$

We see that:

$$\begin{aligned} \mathcal{W}_1(P, Q) &= \sup_{f \in \mathrm{Lip}_1(\mathcal{X}, \mathbb{R})} \mathbb{E}_{x \sim P}[f(x)] - \mathbb{E}_{z \sim Q}[f(z)] \\ &= \inf_{f \in \mathrm{Lip}_1(\mathcal{X}, \mathbb{R})} \mathbb{E}_{x \sim P}[-f(x)] + \mathbb{E}_{z \sim Q}[-(-f(z))] \\ &= \inf_{f \in \mathrm{Lip}_1(\mathcal{X}, \mathbb{R})} \mathbb{E}_{(x,y) \sim \mathbb{P}_{XY}}[\mathcal{L}^W(f(x), y)]. \end{aligned} \quad (19)$$

By Kirszbraun's theorem the optimum of Equation 19 can be extended into a 1-Lipschitz function over $\mathbb{R}^n$. This function can, in turn, be approximated by a LipNet1 network over the domain of interest. $\square$

**Proposition 3. WGAN discriminators are weak classifiers**. *For every $\frac{1}{2} \geq \epsilon > 0$ there exist distributions $P$ and $Q$ with disjoint supports in $\mathbb{R}$ such that for any optimum $f$ of equation 5, the error of classifier sign $\circ f$ is superior to $\frac{1}{2} - \epsilon$.*

*Proof.* We will build $P$ and $Q$ as a finite collection of Diracs. Let $P = \frac{1}{n} \sum_{i=1}^{n} \delta_{4(i-1)}$ and $Q = \frac{1}{n} \sum_{i=1}^{n} \delta_{4i-1}$ for some $n \in \mathbb{N}$, where $\delta_x$ denotes the Dirac distribution in $x \in \mathbb{R}$. A example is depicted in Figure 6 for $n = 20$. In dimension one, the optimal transportation plan is easy to compute: each atom of mass from $P$ at position $i$ is matched with the corresponding one in $Q$ to its immediate right. Consequently we must have $f(4i - 1) = f(4(i - 1)) + 3$. The function $f$ is not uniquely defined on segments $[4i - 1, 4i]$ but it does not matter: since $f$ is 1-Lipschitz we must have $|f(4i-1) - f(4i)| \leq 1$. Consequently in every case for $i < j$ we must have $f(4(i-1)) < f(4(j-1))$ and $f(4i - 1) < f(4j - 1)$. Said otherwise, $f$ is strictly increasing on supp $P$ and supp $Q$.

The solutions of the problems are invariant by translations: if $f$ is the solution, then $f - T$ with $T \in \mathbb{R}$ is also a solution. Let's take a look at classifier $c(x) = \mathrm{sign}(f(x) - T)$. If $T$ is chosen such that $f(4(i - 1)) - T < 0$ and $f(4i - 1) - T > 0$ for some $1 \leq i \leq n$ then $(n - 1) + 2 = n + 1$ points are correctly classified on a total of $2n$ points. It corresponds to an error of $\frac{n-1}{2n} = \frac{1}{2} - \frac{1}{2n}$. Take $n = \lceil \frac{1}{2\epsilon} \rceil$ to conclude. $\square$

## C Proofs of Section 5

### C.1 Consistency of Lipschitz estimators

**Proposition 4. Train Loss is a proxy of Test Loss**. *Let $\mathbb{P}_{XY}$ a probability measure on $\mathcal{X} \times \mathcal{Y}$ where $\mathcal{X} \subset \mathbb{R}^n$ is a bounded set. Let $(x_i, y_i)_{1 \leq i \leq p}$ be a sample of $p$ iid random variables with law $\mathbb{P}_{XY}$. Let $\mathcal{L}$ be a Lipschitz loss function over $\mathbb{R} \times \mathcal{Y}$. We define:*

$$\mathcal{E}_p(f) := \frac{1}{p} \sum_{i=1}^{p} \mathcal{L}(f(x_i), y_i) \text{ and } \mathcal{E}_\infty(f) := \mathbb{E}_{(x,y) \sim \mathbb{P}_{XY}}[\mathcal{L}(f(x), y)]. \quad (7)$$

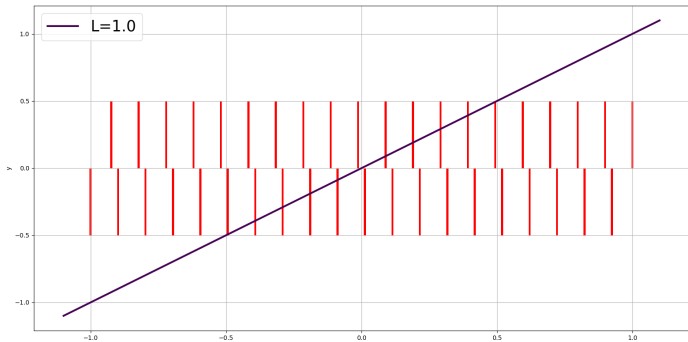

Figure 6: Pathological distributions $P$ and $Q$ of 20 points each, on which the accuracy of the Wasserstein minimizer cannot be better than $52.5\%$.

*Then the empirical loss $\mathcal{E}_p(f)$ converges to the test loss $\mathcal{E}_\infty(f)$ (taking the limit $p \to \infty$):*

$$\min_{f \in Lip_L(\mathcal{X},\mathbb{R})} \mathcal{E}_p(f) \xrightarrow{a.s} \min_{f \in Lip_L(\mathcal{X},\mathbb{R})} \mathcal{E}_\infty(f). \tag{8}$$

**Proof of Theorem 4** is an application of Glivenko-Cantelli theorem.

*Proof.* We proved in Proposition 2 that the minimum of equation 2 is attained, so we replace inf by min for the Lipschitz loss function $\mathcal{L}$. We restrict ourselves to a subset of $\mathrm{Lip}_L(\mathcal{X},\mathbb{R})$ on which $\|f\|_\infty \leq 2L\mathrm{diam}\,\mathcal{X}$ because the minimum lies in this subspace. We have:

$$|\min_f \mathcal{E}_p(f) - \min_f \mathcal{E}_\infty(f)| \leq \max_f |\mathcal{E}_p(f) - \mathcal{E}_\infty(f)|.$$

Let $g_y(x) = \mathcal{L}(f(x), y)$. Note that $g$ is also Lipschitz and bounded on $\mathcal{X}$. The *entropy with bracket* (see [79], Chapter 2.1) of the class of functions $\mathcal{G} = \{g_y = \mathcal{L} \circ f \,|\, f \in \mathrm{Lip}_L(\mathcal{X},\mathbb{R}), y \in \mathcal{Y}, \mathcal{X} \text{ bounded and } \|f\|_\infty \leq 2L\mathrm{diam}\,\mathcal{X}\}$ is finite (see [79], Chapter 3.2). Consequently $\mathcal{G}$ is Glivenko-Cantelli. Finally $\max_f |\mathcal{E}_p(f) - \mathcal{E}_\infty(f)| \xrightarrow{a.s} 0$ which concludes the proof. $\qquad\square$

Results of **Table 1**. Loss $\mathcal{L}_{m,\lambda}^{hkr}$ still belong to Glivenko-Cantelli classes as sum of functions $\mathcal{L}^W$ and $\mathcal{L}_m^H$ from Glivenko-Cantelli classes (on same distribution $\mathbb{P}_X$).

## C.2 Proofs of Section 5.2

**Proposition 5. Optimizing BCE over AllNet leads to divergence**. *Let $f_t$ be a sequence of neural networks, that minimizes the BCE over a non-trivial training set (at least two different examples with different labels) of size $p$, i.e assume that:*

$$\lim_{t \to \infty} \frac{1}{p} \sum_{i=1}^{p} \mathcal{L}_\tau(f_t(x_i), y_i) = 0. \tag{9}$$

*Let $L_t$ be the Lipschitz constant of $f_t$. Then $\lim_{t \to \infty} L_t = +\infty$. There is at least one weight matrix $W$ such that $\lim_{t \to \infty} \|W_t\| = +\infty$. Furthermore, the predicted probabilities are saturated:*

$$\lim_{t \to \infty} \sigma(f_t(x_i)) \in \{0, 1\}. \tag{10}$$

The proof of **Proposition 5** only requires to take a look at the logits of two examples having different labels.

*Proof.* Let $t \in \mathbb{N}$. For the pair $i, j$, as $y_i \neq y_j$, by positivity of $\mathcal{L}$ we must have:

$$0 \leq \mathcal{L}(f_t(x_i), +1) + \mathcal{L}(f_t(x_j), -1) \leq \mathcal{E}(f_t, X). \tag{20}$$

As the right hand side has limit zero, we have:

$$\lim_{t \to \infty} \mathcal{L}(f_t(x_i), +1) = \lim_{t \to \infty} \mathcal{L}(f_t(x_j), -1) = 0$$
$$\implies \lim_{t \to \infty} -f_t(x_i) = \lim_{t \to \infty} f_t(x_j) = -\infty. \tag{21}$$

Consequently $\lim_{t \to \infty} |f_t(x_i) - f_t(x_j)| = +\infty$. By definition $L_t \geq \frac{|f_t(x_i) - f_t(x_j)|}{\|x_i - x_j\|}$ so $\lim_{t \to \infty} L^t = +\infty$. $\qquad\square$

We can always find a network reaching arbitrary small loss on the train set, and arbitrary high loss on the test set. Hence, minimization of train loss does not guarantee generalization.

**Proposition 9** (AllNet networks can always overfit)**.** *Assume that distributions $P$ and $Q$ admit a pdf. Let $n \in \mathbb{N}$, $M > 0$ and $\epsilon > 0$. Let $(x_i, y_i)_{1 \leq i \leq p}$ be a sample of $p$ iid random variables with law $\mathbb{P}_{XY}$ with $x_i \neq x_j$ for all $i \neq j$. Then there exists $f^* \in$ AllNet such that:*

$$f^* \in \{f \in \text{AllNet} \,|\, \mathcal{E}_p(f) = \frac{1}{n} \sum_{i=1}^{n} \mathcal{L}_T(f(x_i), y_i) \leq \epsilon\}$$

*and*

$$\mathcal{E}_\infty(f^*) = \mathbb{E}_{(x,y) \sim \mathbb{P}_{XY}}[\mathcal{L}_T(f^*(x), y)] \geq M.$$

*Proof.* The proof follows the strategy of Proposition 5. Let $d = \min_{\substack{1 \leq i,j \leq n \\ i \neq j}} \|x_i - x_j\|$ the minimum distance between dataset points. We extend the dataset with a new point $(x_{n+1}, y = 1))$ chosen such that $\|x_j - x_{n+1}\| \geq \frac{\delta}{2}$ for all $1 \leq j \leq n$. Then we transform this collections of $n+1$ Diracs functions $\sum_{i=1}^{n+1} \frac{1}{n+1} \delta_{x_i}$ into a a distribution $P$ that admits a pdf by replacing each Dirac with the uniform distribution over the ball of radius $r = \frac{d}{6}$ which yields $P = \sum_{i=1}^{n+1} \frac{1}{n+1} \mathbb{U}(\mathfrak{B}(x_i, r))$. All the balls are disjoint so it exists $f \in$ AllNet such that $\text{sign} f(x_i) = y_i$ for all $1 \leq i \leq n$ and $\text{sign} f(x_{n+1}) = -1$. Now let $|f(x_i)| \to \infty$ to guarantee that $\mathcal{E}_p(f) \to 0$ and $\mathcal{E}_\infty(f) \to \infty$. $\qquad\square$

Fortunately, as soon as the deep learning practitioner restricts itself to a subset of architectures of bounded size, the Proposition 9 is no longer relevant. However, this theorem suggests that if one wants to benefit from useful generalization guarantees, one must keep the architecture of the network fixed once for all while increasing the training set size. This contradicts the trend in deep learning community to use bigger and bigger models when more data becomes available (Resnet-152, GPT3). In the light of this observation, the existence of adversarial attacks should be an expected phenomenon.

Lipschitz networks, on the other side, benefit from Proposition 4: minimization of train loss implies minimization of test loss. Conversely, if the test loss is high and the sample size huge, it means that the train loss is high too.

### C.3   VC dimension for Lipschitz classifiers with margin

We recall below the definition of the Vapnik-Chervonenkis dimension [34] of a class of hypothesis, that build upon shattered sets.

**Definition 9** (**Set shattered by an hypothesis class**)
*Let $\mathcal{Y} = \{-1, +1\}$. Let $\mathcal{H}$ be a class of hypothesis - that is, a set of functions $\mathcal{X} \to \mathcal{Y}$. The set of points $(x_i)_{1 \leq i \leq N} \in \mathcal{X}^N$ is said to be **shattered** by $\mathcal{H}$ if for every sequence of labels $(y_i)_{1 \leq i \leq N} \in \mathcal{Y}^N$, there exists an hypothesis $h \in \mathcal{H}$ such that for every $1 \leq i \leq N$ we have $h(x_i) = y_i$.*

**Definition 10** (**Vapnik-Chervonenkis dimension**)
*The VC dimension of $\mathcal{H}$, denoted $VC_{dim}(\mathcal{H})$, is the greatest integer $N \in \mathbb{N}$ such that it exists a sequence of points $(x_i)_{1 \leq i \leq N} \in \mathcal{X}^N$ shattered by $\mathcal{H}$.*

Roughly speaking, the VC dimension of $\mathcal{H}$ is the size of the biggest set of points such that $\mathcal{H}$ agrees with any label assignment on this set of points. It measures the capacity of a set of classifiers $\mathcal{H}$ to separate some sets of points. The interest of VC dimension introduced in Definition 10 is its link with Probably Approximately Correct (PAC) learning [10].

**Definition 11** (**Agnostic Probably Approximately Correct (PAC) learnability**)
*An hypothesis class $\mathcal{H}$ of functions $\mathcal{X} \to \mathcal{Y}$ is PAC learnable if there exists a function $m_{\mathcal{H}} : (0,1)^2 \to \mathbb{N}$ and a learning algorithm $\mathcal{D} \mapsto h_m$ such that for every $(e, \beta) \in (0,1)^2$, for any distribution $\mathbb{P}_{XY}$ on $\mathcal{X} \times \mathcal{Y}$, for any dataset $\mathcal{D} = ((x_1, y_1), (x_2, y_2), \ldots, (x_m, y_m)) \overset{iid}{\sim} \mathbb{P}_{XY}$ of size $m \geq m_{\mathcal{H}}(e, \beta)$, we have:*

$$\mathbb{P}(E_{\mathbb{P}_{XY}}(h_m) \leq \min_{h \in \mathcal{H}} E_{\mathbb{P}_{XY}}(h) + e) \geq 1 - \beta$$

*We denote by $E_{\mathbb{P}_{XY}}(h) := \mathbb{E}_{(x,y) \sim \mathbb{P}_{XY}}[\mathbb{1}\{h(x) \neq y\}]$ the empirical risk: the expectation of error function over $\mathbb{P}_{XY}$.*

Roughly speaking, for an agnostic PAC learnable class, the probability to pick the best hypothesis $h^* \in \mathcal{H}$ up to error $e > 0$ happens with probability at least $1 - \beta > 0$ over datasets of size at least $m_{\mathcal{H}}(e, \beta)$ sampled from distribution $\mathbb{P}_{XY}$. This definition captures the hypothesis classes that are "small enough" such that a reasonably high number of samples allows you to pick the best hypothesis by high probability.

The implication "finite VC dimension" $\implies$ "agnostic PAC learnable" is a classical result from [80]. This motivates to compute the VC dimension of Lipschitz classifiers: it yields PAC learnability results.

**Proposition 6. 1-Lipschitz Functions with margin are PAC learnable**. *Assume $P$ and $Q$ have bounded support $\mathcal{X}$. Let $m > 0$ the margin. Let $\mathcal{C}^m(\mathcal{X}) = \{c_f^m : \mathcal{X} \to \{-1, \bot, +1\}, f \in Lip_1(\mathcal{X}, \mathbb{R})\}$ be the hypothesis class defined as follow.*

$$c_f^m(x) = \begin{cases} +1 & \text{if } f(x) \geq m, \\ -1 & \text{if } f(x) \leq -m, \\ \bot & \text{otherwise, meaning "} f \text{ doesn't feel confident".} \end{cases} \tag{11}$$

*Let $\mathfrak{B}$ be the unit ball. Then the VC dimension of $\mathcal{C}^m$ is finite:*

$$(\frac{1}{m})^n \frac{vol(\mathcal{X})}{vol(\mathfrak{B})} \leq VC_{dim}(\mathcal{C}^m(\mathcal{X})) \leq (\frac{3}{m})^n \frac{vol(\mathcal{X})}{vol(\mathfrak{B})}. \tag{12}$$

*Proof.* This approach with margins $m$ yields objects known in the literature as $m$-fat shattering sets [71].

The VC dimension of $\mathcal{C}^m(\mathcal{X})$ is the maximum size of a set shattered by $\mathcal{C}^m(\mathcal{X})$. As the functions $f$ are 1-Lipschitz, if $c_f^m(x) = -c_f^m(y)$ then $f(x) \geq m$, $f(y) \leq m$ and $\|x - y\| \geq 2m$. Consequently, a finite set $X \subset \mathcal{X}^n$ is shattered by $\mathcal{C}^m(\mathcal{X})$ if and only if for all $x, y \in X$ we have $\mathfrak{B}(x, m) \cap \mathfrak{B}(y, m) = \varnothing$ where $\mathfrak{B}(x, m)$ is the open ball of center $x$ and radius $m$.

The maximum number of disjoint balls of radius $m$ that fit inside $\mathcal{X}$ is known as the **packing number** of $\mathcal{X}$ with radius $m$. $\mathcal{X}$ is bounded, hence its packing number is finite.

The bounds on the packing number are a direct application of [81] (Lemma 1). $\qquad\square$

### C.4 VC dimension for GroupSort networks

With GroupSort2 activation functions (as in the work of [36]) we get the following rough upper bound:

**Proposition 7. VC dimension of LipNet1 neural networks**. *Let $f_\theta : \mathbb{R}^n \to \mathbb{R}$ a LipNet1 neural network with parameters $\theta \in \Theta$, with **GroupSort2** activation functions, and a total of $W$ neurons. Let $\mathcal{H} = \{sign f_\theta | \theta \in \Theta\}$ the hypothesis class spanned by this architecture. Then we have:*

$$VC_{dim}(\mathcal{H}) = \mathcal{O}\left((n+1)2^W\right). \tag{13}$$

From Proposition 7 we can derive generalization bounds using PAC theory. Note that most results on VC dimension of neural network use the hypothesis that the activation function is applied element-wise (such as in [35]) and get asymptotically tighter lower bounds for ReLU case. Such hypothesis does not apply anymore here, however we believe that this preliminary result can be strengthened. Our result is actually a bit more general and applies more broadly to activation functions that piece-wise linear and partition the input space into convex sets.

**The proof of Proposition 7** uses the number of affine pieces generated by GroupSort2 activation function, and the VC dimension of piecewise affine classifiers with convex regions.

*Proof.* First, we need the following lemma.

**Lemma 1** (Piecewise affine function). *Let $\mathcal{H}$ a class of classifiers that are piecewise affine, such that the pieces form a convex partition of $\mathbb{R}^n$ with $B$ pieces (each piece of the partition is a convex set). Then we have:*

$$VC_{dim}(\mathcal{H}) = \mathcal{O}\left((n+1)B^2\right).$$

The proof of Lemma 1 is detailed below.

Let $\mathcal{G}(N)$ be the **growth function** [82] of $\mathcal{H}$. According to Sauer's lemma [82] if it grows polynomially with the number of points, then the degree of the polynomial is an upper bound on the VC dimension. We will show that is indeed the case by computing a crude upper bound of the degree. Assume that we are given $N$ points, and $N$ big enough such that Sauer's lemma can be applied.

Assume that we can choose freely the convex partition, and then only the affine classifier inside each piece. In general for neural networks that might not be the case (the boundary between partitions depends of the affine functions inside it, since neural networks are continuous); however, we are only interested in an upper bound so we can consider this generalization.

Each piece of the partition is a polytope [83]. Each polytope is characterized by a set of exactly $B-1$ affine inequalities since each polytope is the intersection of $B-1$ halfspaces [83]. The whole partition is characterized by $\frac{B(B-1)}{2}$ affine inequalities. We divide by two because of the symmetry. Hence there exists an injective map from the set of convex partitions with $B$ pieces into $(\mathbb{R}^{n+1})^{\frac{B(B-1)}{2}}$. It is not a bijective map in general, since different systems might describe the same partition, and some degenerate systems do not correspond to partitions at all.

We split the problem and consider each one of the $\frac{B(B-1)}{2}$ inequalities independently. According to Sauer's lemma, there is $\mathcal{O}(N^{n+1})$ ways to place the first hyperplane characterizing the first halfspace. Idem for the second hyperplane, and so on. Hence, there is at most $\mathcal{O}((N^{n+1})^{\frac{B(B-1)}{2}})$ ways to assign the $N$ points to the $B$ convex bodies.

Each convex body (among the $B$ of them) contains atmost $N$ points, on which (still according to Sauer's lemma) there is at most $\mathcal{O}(N^{n+1})$ way to assign them labels $+1$ or $-1$, since the classifier is piecewise **affine**.

Consequently, we have $\mathcal{G}(N) = \mathcal{O}((N^{n+1})^{\frac{B(B-1)}{2}}(N^{n+1})^B) = \mathcal{O}((N^{n+1})^{\frac{B(B+1)}{2}}) = \mathcal{O}((N^{n+1})^{B^2})$. Sauer's lemma allows us to conclude:

$$VC_{\dim}(\mathcal{H}) = \mathcal{O}\left((n+1)B^2\right).$$

**Proof of the result.** Now, we need to prove that $f$ is piecewise affine and the number of such pieces is not greater than $\prod_{i=1}^{k} 2^{\frac{w_i}{2}} = \sqrt{2^W}$, where $w_i$ is the number of neurons in layer $i$. We proceed by induction on the depth of the neural network. For depth $K = 0$ we have an affine function $\mathbb{R}^n \to \mathbb{R}$ which contains only one affine piece by definition (the whole domain), so the result is true.

Now assume that a neural network $\mathbb{R}^{w_1} \to \mathbb{R}$ of depth $K$ with widths $w_2 w_3 \ldots w_k$ has $S_k$ affine pieces. The enumeration starting at $w_2$ is not a mistake: we pursue the induction for a neural network $\mathbb{R}^n \to \mathbb{R}$ of depth $K+1$ and widths $w_1 w_2 \ldots w_k$. The composition of affine function is affine, hence applying an affine transformation $\mathbb{R}^n \to \mathbb{R}^{w_1}$ preserves the number of pieces. The analysis falls back to the number of distinct affine pieces created by GroupSort2 activation function. If such activation function creates $S$ pieces then we have the immediate bound $S_{K+1} \leq SS_k$.

Let $(Jf)(x) \in \mathbb{R}^{w_1 \times w_1}$ be the Jacobian of the GroupSort2 operation evaluated in $x$. The cardinal $|\{(Jf)(x), x \in \mathbb{R}^{w_1}\}|$ is the number of distinct affine pieces. For GroupSort2 we have combinations of $\frac{w_i}{2}$ MinMax gates. Each MinMax gate is defined on $\mathbb{R}^2$ and contains two pieces: one on which the gate behaves like identity and the other one on which the gate behaves like a transposition. Consequently we have $S_{k+1} \leq 2^{\frac{w_k}{2}} S_k$ and unrolling the recurrence yields the desired result.

Finally, we just need to apply the Lemma 1 with $B = \sqrt{2^W}$.

□

## C.5 Generalization bounds literature survey

In [70] a link is established between Lipschitz classifiers and linear large-margin classifiers. Generalization bounds for large class of Lipschitz classifiers are provided by the work of [71] using Vapnik–Chervonenkis theory. Other generalization bounds related to spectral normalization can be found in [72]. Links between adversarial robustness, large margins classifiers and optimization bias are studied in [75, 73, 74]. The importance of the loss in adversarial robustness is studied in [76]. In [16], the control of Lipschitz constant and margins is used to guarantee robustness against attacks. A link between classification and optimal transport is established in [8] by considering a hinge regularized version of the Kantorovich-Rubinstein dual objective.

## D   Deel.Lip networks

The theorem 3 of [11] bound the $\| \cdot \|_{2 \to \infty}$ [12] and $\| \cdot \|_\infty$ norms of weight matrices to obtain universal approximation in $\mathrm{Lip}_1(\mathcal{X}, \mathbb{R})$ . In practice, they reported that bounding spectral norm $\| \cdot \|_2$ and enforcing orthogonality of rows/columns of weight matrices yielded the best empirical results, because it turned the network into a *Gradient Norm Preserving* network. Unfortunately, this last construction still lacks universal approximation results. Nonetheless, neither [11] nor ourselves encountered (so far) a function that couldn't be approximated by those GNP networks.

All the experiments done in the paper use the Deel-Lip[3] library [8], following ideas of [11]. The networks use 1) orthogonal matrices and 2) GroupSort2 activation. Orthogonalization is enforced using Spectral normalization [14] and Björck algorithm [15]. We have for all $i$:

$$\mathrm{GroupSort2}(x)_{2i,2i+1} = [\min(x_{2i}, x_{2i+1}), \max(x_{2i}, x_{2i+1})].$$

The networks parameterized by this library are GNP and belong to LipNet1  by construction.

The implementation of Lipschitz neural networks benefits from a rich literature. We outline below the most significant results and contributions of literature, that motivated us to use Deel-Lip library.

**LipNet1 networks parametrization.**   The Lipschitz constant of affine layers can be constrained with a Gradient penalty [37] (WGAN) or spectral regularization [38], without formal guarantee, only a very crude upper bound. Weight clipping [6] (WGAN), Frobenius normalization [39] and spectral normalization [14] lead to a tighter upper bound. However, naively stacking such layers leads to vanishing gradients. Most activation functions are Lipschitz, the popular including ReLU, sigmoid, tanh, softplus; layers such as Attention are not Lipschitz [40]. Lipschitz recurrent units have been proposed in [41, 42]. Residual connections are Lipschitz but prone to vanishing gradients (see Appendix F).

**Gradient Norm Preserving networks.**   In [37], authors show that the potential $f$ of the Kantorovich-Rubinstein dual transport problem verifies $\|\nabla_x f(x)\| = 1$ almost everywhere on the support of the distributions $\mathbb{P}_{XY}$. LipNet1 networks fulfilling $\|\nabla_x f(x)\| = 1$ almost everywhere wrt any intermediate activation $x$ are said to be Gradient Norm Preserving (GNP), and elegantly avoids the vanishing gradients phenomenon [3, 45]. This is typically achieved in affine layers with orthogonal matrices, which justify the *"orthogonal neural network"* terminology [46, 47]. [11] establish that GNP networks with ReLU are exactly affine functions. They proposed Sorting activation functions to circumvent this expressiveness issue. In particular GroupSort2 revealed to be an efficient alternative [36] to ReLU, and can be seen as a particular case of Householder reflections [49, 50]. Other authors tried to fix ReLU itself [43].

**Orthogonal kernels**   are of special interest in the context of normalizing flows [58], ensemble methods [59], reinforcement learning [60] or graph neural networks [61]. The optimization over the orthogonal group (known as Stiefel manifold) has been extensively studied in [62], while  [63, 64, 64, 65, 66] focus on neural networks retractions like Cayley transform or exponential map; more recently [67] proposed a landing algorithm, [68] proposed an algorithm inspired by quantum

---

[3]`https://github.com/deel-ai/deel-lip` distributed under MIT License (MIT).

computing, and [69] proposed an approach based on graph matching. Orthogonal convolutions are still an active research area: the constraint is enforced by using appropriate regularization [51, 52], by expressing convolutions in Fourrier space [53, 54], or by optimizing over the set of orthogonal convolutions directly [3, 55, 56].

In order to build CNN we used the convolution layers already provided in Deel-Lip. One limitation of these layers is that it uses the Reshaped Kernel Orthogonalization (RKO) [3] method which, although it ensures Lipschitz bounds, does not guarantee exact orthogonality.

We also attempted to use Skew Orthogonal Convolutions (SOC, as described in algorithm 1 of [56]). However, when we performed a sanity-check with the power iteration method, we obtained convolutions with Lipschitz constant greater than 1.

As the method used to build 1-Lipschitz networks does not affect the conclusions of our paper, we decided to stick with Lipschitz constant provably smaller than one. We did not observe improvements by using Householder activation functions, so we used GroupSort2 activation functions instead (which are computationally cheaper).

## E  Multiclass Hinge Kantorovich Rubinstein

The loss HKR proposed by [8] was originally designed for binary classification. There are several ways to adapt it to the multi-class $K > 2$ setting.

The most obvious one would be a *one-versus-all* scheme. However, in multiclass classification the prediction is given by $\arg\max f_k$ and not by $\text{sign} \circ f$, so $f^{-1}(\{0\})$ is not longer the frontier. Consequently, this approach fails to yield meaningful certificates.

Instead the construction of HKR loss should once again rely on Multiclass Mean Certificate Robustness (see Definition 12). Indeed, the robustness radius $\delta$ for class $k$ verifies:

$$\|\delta\| \geq \|f(x+\delta) - f(x)\| \geq \frac{1}{2}\left(f_k(x) - \max_{i \neq k} f_i(x)\right).$$

The $\frac{1}{2}$ comes from the fact that each $f_i$ is 1-Lipschitz, so their difference is 2-Lipschitz at most. This definition is coherent with the one of *multiclass hinge loss* found in most frameworks. We compare the logits of the true class with the ones of the closest other class to weight the certificate positively or negatively according to the true label.

**Definition 12** (Multiclass Mean Certifiable Robustness (MMCR))
*For any function $f : \mathcal{X} \to \mathbb{R} \in LipNet1$ we define its weighted multiclass mean certifiable robustness $\mathcal{R}_{(P,y)}(f)$ on class $P$ with label $k$ as:*

$$\mathcal{R}_{(P,y)}(f) := \mathbb{E}_{x \sim P}[f_k(x) - \max_{i \neq k} f_i(x)]. \tag{22}$$

*Note that $f_k(x) - \arg\max_{i \neq k} f_i(x)$ is either positive or negative, according to the prediction.*

Then we define the Multiclass HKR:

**Definition 13** (Multiclass HKR)
*For class an example $x$ of label $k$ let:*

$$R_k(x) := f_k(x) - \arg\max_{i \neq k} f_i(x).$$

*We define the multiclass HKR as:*

$$\mathcal{L}_\lambda^M(f(x), k) := -R_k(x) + \alpha \max(0, m - R_k(x)).$$

For $K = 2$ we recover the binary case on the function $\hat{f} = f_1 - f_2$. Experiments showed that the *one versus all* approach was outperformed by the multiclass HKR, in both robust accuracy and training time.

All $f_k$ functions can be learned independently, however in practice they share the same Lipschitz backbone and only differ in the last layer, as early experiments showed that it did not impact negatively the results. Using the same arguments as Proposition 2 based on Arzelà-Ascoli theorem we show that the minimum of $\mathcal{L}_\lambda^M$ is well defined and attained for each $f_k$.

# F Gradient Norm Preserving (GNP) networks

Vanishing and Exploding gradients have been a long-time issue in the training of neural networks. The latter is usually avoided by regularizing the weights of the networks and using bounded losses, while the former can be avoided using residual connections (such ideas can found on LSTM [84] or ResNet [85]). On Gradient Norm Preserving (GNP) networks (orthogonal networks with GroupSort activation such as the ones of *Deel.lip* library), we can guarantee the absence of exploding gradient:

**Proposition 10** (No exploding gradients [3])**.** *Assume that $f = h^M \circ h^{M-1} \circ \ldots \circ h^2 \circ h^1$ is a feed-forward neural network and that each layer $h^i$ is 1-Lipschitz, where $h^i$ is either a 1-Lipschitz affine transformation $h^i(x) = W^i x + B^i$ either a 1-Lipschitz activation function. Let $\mathcal{L} : \mathbb{R}^k \times \mathcal{Y} \to \mathbb{R}$ the loss function. Let $\tilde{y} = f(x)$, $H^i = h^i \circ h^{i-1} \circ \ldots \circ h^2 \circ h^1$ and $H^0(x) = x$. Then we have:*

$$\|\nabla_{W^i}\mathcal{L}(\tilde{y},y)\| \leq \|\nabla_{\tilde{y}}\mathcal{L}(\tilde{y},y)\|\|H^{i-1}(x)\| \tag{23}$$

$$\|\nabla_{B^i}\mathcal{L}(\tilde{y},y)\| \leq \|\nabla_{\tilde{y}}\mathcal{L}(\tilde{y},y)\|. \tag{24}$$

To prove **Proposition 10** we just need to write the chain rule.

*Proof.* The gradient is computed using chain rule. Let $\theta$ be any parameter of layer $h^i$. Let $h^j_\perp$ be a dummy variable corresponding to the input of layer $h^j$, which is also the output of layer $h^{j-1}$. Then we have:

$$\nabla_\theta \mathcal{L}(\tilde{y},y) = \nabla_{\tilde{y}}\mathcal{L}(\tilde{y},y) M (J_\theta h^j(H^{i-1}(x))). \tag{25}$$

with $M = \left(\prod_{j=M}^{i+1} J_{h^j_\perp} h^j(H^{j-1}(x))\right)$. As the layers of the neural network are all 1-Lipschitz, we have:

$$\|J_{h^j_\perp} h^j(H^{j-1}(x))\| \leq 1.$$

Hence we get the following inequality:

$$\|\nabla_\theta \mathcal{L}(\tilde{y},y)\| \leq \|\nabla_{\tilde{y}}\mathcal{L}(\tilde{y},y)\|\|J_\theta h^j(H^{i-1}(x))\|. \tag{26}$$

Finally, for $h^i(H^{i-1}(x)) = W^i H^{i-1}(x) + B^i$ we replace $\theta$ by the appropriate parameter which yields the desired result. $\square$

There is still a risk of vanishing gradient, which strongly depends of the loss $\mathcal{L}$. For Lipschitz neural networks, BCE $\mathcal{L}_T$ does not suffer from vanishing gradient.

**Proposition 11** (No vanishing BCE gradients)**.** *Let $(x_i, y_i)_{1\leq i\leq p}$ be a non trivial training set (i.e with more than one class) such that $x_i \in \mathcal{X}$, $\mathcal{X}$ a **bounded** subset of $\mathbb{R}^n$. Then there exists a constant $K > 0$ such that, for every minimizer $f_L^*$ of BCE (known to exist thanks to Proposition 2) we have:*

$$f_L^* \in \arg\min_{f \in Lip_L(\mathcal{X},\mathbb{R})} \mathbb{E}_{(x,y)\sim\mathbb{P}_{XY}}[\mathcal{L}_T^{bce}(f(x),y)]. \tag{27}$$

*And such that for every $1 \leq i \leq p$ we have the following:*

$$|\frac{\partial}{\partial\tilde{y}}\mathcal{L}_T^{bce}(\tilde{y} = f_L^*(x_i), y_i)| \geq K. \tag{28}$$

*Note that $K$ only depends of the training set, not $f_L^*$.*

*Proof.* Note that it exists $K' > 0$ such that $|f_L^*(x_i)| \leq K'$ for all $x_i$ and all minimizers $f_L^*$, just like in the proof of Proposition 2, because otherwise we could exhibit a sequence of minimizers $(f_L^*)_t$ not uniformly bounded, which is a contradiction.

Consequently $|\frac{\partial}{\partial\tilde{y}}\mathcal{L}_T^{bce}(\tilde{y} = f(x_i), y_i)| \geq \frac{1}{1+\exp(|f(x_i)|)} \geq \frac{1}{1+\exp(K')} = K$. $\square$

It means that a non-null gradient will remain for each training example taken independently, but their mean over the train set after convergence will be the null vector. Consequently, we must expect high variance in gradients and oscillations when we get closer to the minimum.

We used *VGG-like* architectures instead of Resnet because GNP property makes residuals connections less useful overall (no need for shortcuts when gradient is preserved), and because those residual connections can actually be harmful:

**Remark** (Residual connections). *If $f$ verifies $\|\nabla_x f(x)\| = 1$ almost everywhere, and if $g$ verifies $\|\nabla_x g(x)\| = 1$ almost everywhere, then $\|\nabla_x(\frac{1}{2}f(x) + \frac{1}{2}g(x))\| < 1$ in general, unless $\nabla_x f(x) = \nabla_x g(x)$. Taking $f(x) = x$ we end up with residual connections, for which ensuring $\|\nabla_x(\frac{1}{2}f(x) + \frac{1}{2}g(x))\| = 1$ almost everywhere is not possible unless $f = g$.*

Remark essentially shows that the set of GNP layers is not stable by sum or other common operations. This makes their practical implementation and the demonstration of universal approximation theorems trickier.

## G  Wasserstein discriminator does not depend of the Lipschitz constant

The dual problem can be reformulated by swapping the objective and the constraint:

$$
\begin{aligned}
\arg \min_{Pf-Qf \geq \epsilon \mathcal{W}(P,Q)} \operatorname{Lip}(f) &= \epsilon \arg \min_{Pf-Qf \geq \mathcal{W}(P,Q)} \operatorname{Lip}(f) \\
&= \epsilon \arg \max_{\operatorname{Lip}(f)=1} Pf - Qf \\
&= \arg \max_{\operatorname{Lip}(f)=\epsilon} Pf - Qf.
\end{aligned}
\tag{29}
$$

$\epsilon$ can be seen as re-scaling (change of units in physicist vocabulary). This makes more clear the fact that changing the Lipschitz constant is just changing the units used to measure distance. The invariance by dilation mentioned in Section 4 must be understood in this sense: any constant $L$ can be chosen for the computation of $\mathcal{W}_1$ as long as this constant is chosen in advance and bounded throughout the optimization process.

## H  BCE through the lens of OT

In the following, we try to draw links between BCE minimization and optimal transport. Since the objective function is optimized with gradient descent, the gradients of the loss is the object of interest. We re-introduce $f_\theta$ as a function parameterized by $\theta$, mapping the input to the logits. Let $g_\theta^p(x) = \sigma(f_\theta(x))$ and $g_\theta^q(x) = 1 - \sigma(f_\theta(x))$. $g_\theta^p(x)$ (resp. $g_\theta^q(x)$) are the predicted probabilities of class $+1$ (resp. -1).

Now define $\mathcal{Z}_\theta^p = \mathbb{E}_{x \sim P}[g_\theta^q(x)]$ and $\mathcal{Z}_\theta^q = \mathbb{E}_{x \sim Q}[g_\theta^p(x)]$. $\mathcal{Z}_\theta^p$ can be seen as the weighted rate of false negatives. That is, the average mass of probability given to class $-1$ by $f_\theta$ when examples are sampled from class $+1$. Similarly, $\mathcal{Z}_\theta^q$ can be seen as the rate of false positives. Let:

$$
\mathrm{d}P_\theta(x) = \frac{1}{\mathcal{Z}_\theta^p} g_\theta^q(x)\,\mathrm{d}P(x) \text{ and } \mathrm{d}Q_\theta(x) = \frac{1}{\mathcal{Z}_\theta^q} g_\theta^p(x)\,\mathrm{d}Q(x).
\tag{30}
$$

Consequently, $P_\theta$ (resp. $Q_\theta$) is a valid probability distribution on $\mathbb{R}^n$ corresponding to the probability of an example $x$ to be incorrectly classified in class $-1$ (resp. $+1$). With these notations, the full expression of the gradient takes a simple form. Behold the minus sign: it is a gradient *descent* and not a gradient *ascent*.

$$
-\nabla_\theta \left(\mathbb{E}_{x \sim P}[\mathcal{L}(f_\theta(x), +1)] + \mathbb{E}_{x \sim Q}[\mathcal{L}(f_\theta(x), -1)]\right) = \mathcal{Z}_\theta^p \mathbb{E}_{x \sim P_\theta}[\nabla_\theta f_\theta(x)] - \mathcal{Z}_\theta^q \mathbb{E}_{x \sim Q_\theta}[\nabla_\theta f_\theta(x)]
\tag{31}
$$

We apply a bias term $T \in \mathbb{R}$ to classify with $f_\theta - T$ instead. For a well-chosen $T$ we can enforce $\mathcal{Z}_\theta^p = \mathcal{Z}_\theta^q$, and such $T$ can be found using the bisection method. The optimization is performed over the set of 1-Lipschitz functions. We end up with:

$$
\mathcal{Z}_\theta^p (\mathbb{E}_{x \sim P_\theta}[\nabla_\theta f_\theta(x)] - \mathbb{E}_{x \sim Q_\theta}[\nabla_\theta f_\theta(x)]).
\tag{32}
$$

This is the gradient for the computation of Wasserstein metric $\mathcal{W}$ between $P_\theta$ and $Q_\theta$, using Rubinstein-Kantorovich dual formulation. Hence, binary cross-entropy minimization is similar to the computation of a transportation plan between errors distributions $P_\theta$ and $Q_\theta$. Note that $P_\theta$ and $Q_\theta$ depend of the current classifier $f_\theta - T$, so the problem is not stationary.

Finally, observe that $\mathcal{L}_\tau^{bce}(f(x), y) = \log 2 - \frac{y\tau f(x)}{2} + \mathcal{O}(\tau^2 f^2(x))$ so when $\tau \to 0$ we get:

$$
\min_{f \in \operatorname{Lip}_1(\mathcal{X}, \mathbb{R})} \frac{4}{\tau} \left(\mathbb{E}_{(x,y) \sim P_{XY}}[\mathcal{L}_\tau^{bce}(f(x), y)] - \log 2\right) = -\mathcal{W}_1(P, Q).
$$

| parameter | value |
|---|---|
| data augmentation | none |
| input scale | $[0, 1]$ |
| batch size | 1000 |
| learning rate | 0.001 |
| optimizer | Adam |
| cosine decay | 0.01 |
| architecture | 1024-1024-100 |
| activation | GroupSort2 |
| epochs | 250 |

(a) Learning parameters on CIFAR-100 with random labels.

| loss | $CCE$ | $HKR$ |
|---|---|---|
| | $\tau = 256$ | $\alpha = 256$ |
| | | $m = \frac{36}{255}$ |
| accuracy | 0.999 | 0.998 |
| robust accuracy $\epsilon = 36$ | 0.382 | 0.91 |
| robust accuracy $\epsilon = 72$ | 0.021 | 0.19 |
| lipschitz upper bound | 1.002 | 1.002 |

(b) Learning results for CIFAR-100 with random labels: validation accuracy is not reported as its value is always 0.01 for clean accuracy and 0.00 for robust accuracy. The Lipschitz upper bound is computed using the power iteration method on each layer.

Figure 7

In the limit of small temperatures, the Binary Cross-Entropy is essentially equivalent to Wasserstein. In AllNet networks, as the training proceeds, the Lipschitz constant increases (equivalently increasing $\tau$) and the loss self-correct with $P_\theta$ and $Q_\theta$ to improve accuracy.

# I Fitting CIFAR100 with random labels

This experiment illustrates that constraining the Lipschitz of a network does not affect its expressive power. To show this we train a constrained network on the CIFAR100 dataset where all labels have been replaced with random labels, this task is now a widely recognized benchmark to evaluate the expressiveness of an architecture [86].

The architecture of this network is as simple as possible: two orthogonal dense layers with 1024 neurons are followed by a dense layer which is normalized but not orthogonal. The GroupSort2 activation function is used and biases are enabled.

Hyper parameters for this experiment are listed in table 7a, and results are reported in table 7b.

At first glance it's might seem surprising to see both high accuracy and high provable robustness on a dataset with random labels. This is compliant with the idea expresses by the authors of [24]: for a given accuracy one can increase the robustness radius around a sample $x_1$ up to the value $\|\frac{x_1 - x_2}{2}\|$ where $x_2$ is the closest sample with a different label. The decision frontier is close to the decision frontier of the 1-nearest neighbor based on the trained set. This illustrates that constraining the Lipschitz constant does not necessarily decrease accuracy and does not necessarily increase robustness. Also, it shows that there is no trivial link between robustness and generalization.

# J 1-Lipschitz estimators are consistent (experimental protocol of figure 4)

Lipschitz classifiers are consistent: as the size of the training set increases, the training loss becomes a proxy for the test loss. However, we do not give convergence speed bounds: we do not know how many samples are needed for a given task to observe the convergence between train and test losses. Moreover, the losses are parametrized (e.g by $\tau, \alpha, m$) so we expect to have different convergence rates, depending on those parameters. In order to observe this empirically on the CIFAR10 dataset, the same architecture (described in fig 9a) was trained successively on 2%, 5%, 10%, 25%, 50% and 100% of the dataset. The sub-sampling was performed with a different seed each time, showing that, for this range of $\tau$ training is still stable. Similarly, this procedure has been repeated with different values for $\tau$. Hyper-parameters used for learning are reported in fig 9b. Learning results are reported in Figure 8. It also shows certifiable accuracy and empirical accuracy, that were not displayed in the fig 4 from the main paper. We see that lower values for $\tau$ yield tighter robustness certificates (certificate value is close to the distance found by L2-PGD).

| $\tau$ | % dataset | accuracy | certifiable $\epsilon:36$ | L2-PGD $\epsilon:36$ | certifiable $\epsilon:72$ | L2-PGD $\epsilon:72$ | certifiable $\epsilon:108$ | L2-PGD $\epsilon:108$ |
|---|---|---|---|---|---|---|---|---|
| 8.0 | 1 | 0.6279 | 0.3075 | 0.525 | 0.0996 | 0.462 | 0.0204 | 0.388 |
| 4.0 | 1 | 0.6207 | 0.4289 | 0.568 | 0.2553 | 0.502 | 0.1308 | 0.443 |
| 2.0 | 1 | 0.5683 | 0.4456 | 0.535 | 0.3298 | 0.496 | 0.2313 | 0.457 |
| 1.0 | 1 | 0.5097 | 0.4331 | 0.482 | 0.3615 | 0.436 | 0.2926 | 0.404 |
| 0.5 | 1 | 0.4497 | 0.398 | 0.434 | 0.3478 | 0.4 | 0.3035 | 0.369 |
| 0.25 | 1 | 0.4064 | 0.3703 | 0.395 | 0.3346 | 0.373 | 0.2974 | 0.351 |
| 8.0 | 0.5 | 0.5959 | 0.2862 | 0.526 | 0.0973 | 0.443 | 0.0218 | 0.363 |
| 4.0 | 0.5 | 0.5884 | 0.3967 | 0.511 | 0.2275 | 0.44 | 0.1123 | 0.383 |
| 2.0 | 0.5 | 0.5569 | 0.4393 | 0.503 | 0.3298 | 0.463 | 0.2277 | 0.419 |
| 1.0 | 0.5 | 0.5012 | 0.4235 | 0.443 | 0.3525 | 0.403 | 0.2875 | 0.366 |
| 0.5 | 0.5 | 0.4486 | 0.3996 | 0.419 | 0.3517 | 0.397 | 0.3093 | 0.367 |
| 0.25 | 0.5 | 0.3928 | 0.3574 | 0.373 | 0.3224 | 0.352 | 0.2969 | 0.331 |
| 8.0 | 0.25 | 0.5553 | 0.2801 | 0.484 | 0.0987 | 0.402 | 0.0252 | 0.322 |
| 4.0 | 0.25 | 0.5696 | 0.3835 | 0.493 | 0.2208 | 0.437 | 0.112 | 0.382 |
| 2.0 | 0.25 | 0.5397 | 0.4156 | 0.482 | 0.299 | 0.431 | 0.2059 | 0.384 |
| 1.0 | 0.25 | 0.5013 | 0.4205 | 0.455 | 0.345 | 0.423 | 0.2779 | 0.389 |
| 0.5 | 0.25 | 0.4448 | 0.3919 | 0.418 | 0.3449 | 0.392 | 0.2983 | 0.372 |
| 0.25 | 0.25 | 0.3939 | 0.3593 | 0.359 | 0.3278 | 0.348 | 0.2944 | 0.327 |
| 8.0 | 0.1 | 0.4914 | 0.2617 | 0.396 | 0.1133 | 0.335 | 0.0396 | 0.271 |
| 4.0 | 0.1 | 0.5053 | 0.3397 | 0.449 | 0.2001 | 0.378 | 0.1024 | 0.322 |
| 2.0 | 0.1 | 0.503 | 0.3858 | 0.478 | 0.2793 | 0.426 | 0.1898 | 0.387 |
| 1.0 | 0.1 | 0.4783 | 0.3977 | 0.428 | 0.3188 | 0.391 | 0.2484 | 0.355 |
| 0.5 | 0.1 | 0.4385 | 0.3824 | 0.425 | 0.331 | 0.401 | 0.2799 | 0.383 |
| 0.25 | 0.1 | 0.3872 | 0.3517 | 0.379 | 0.3168 | 0.35 | 0.2874 | 0.328 |
| 8.0 | 0.05 | 0.445 | 0.266 | 0.406 | 0.1324 | 0.337 | 0.0585 | 0.264 |
| 4.0 | 0.05 | 0.4508 | 0.3052 | 0.439 | 0.187 | 0.374 | 0.1078 | 0.309 |
| 2.0 | 0.05 | 0.4562 | 0.3444 | 0.395 | 0.2478 | 0.342 | 0.1723 | 0.293 |
| 1.0 | 0.05 | 0.4392 | 0.3579 | 0.416 | 0.2898 | 0.377 | 0.2272 | 0.343 |
| 0.5 | 0.05 | 0.4176 | 0.3629 | 0.396 | 0.3106 | 0.365 | 0.2679 | 0.326 |
| 0.25 | 0.05 | 0.3741 | 0.3379 | 0.352 | 0.3028 | 0.323 | 0.2707 | 0.306 |
| 8.0 | 0.02 | 0.3778 | 0.2539 | 0.327 | 0.159 | 0.27 | 0.0922 | 0.227 |
| 4.0 | 0.02 | 0.3761 | 0.2738 | 0.298 | 0.1879 | 0.254 | 0.1236 | 0.207 |
| 2.0 | 0.02 | 0.3859 | 0.2965 | 0.325 | 0.2167 | 0.278 | 0.1549 | 0.236 |
| 1.0 | 0.02 | 0.385 | 0.3129 | 0.35 | 0.2522 | 0.312 | 0.1999 | 0.273 |
| 0.25 | 0.02 | 0.3378 | 0.3021 | 0.305 | 0.2684 | 0.288 | 0.2375 | 0.26 |
| 0.5 | 0.02 | 0.3631 | 0.3091 | 0.322 | 0.2605 | 0.302 | 0.2172 | 0.282 |

Figure 8: Network trained on different fractions of the CIFAR-10 dataset. For each value of $\tau$ and each dataset fraction, clean accuracy, certifiable and empirical accuracies are reported. We report accuracy under l2-PGD attack to perform a sanity check of the network's certificates. Interestingly, a lower temperature leads to a tighter bound for certifiable robustness (a lower gap between certifiable robustness and empirical robustness).

| network architecture |
|---|
| conv-3x3-32 (groupsort 2) |
| conv-3x3-32 (groupsort 2) |
| invertible downsampling |
| conv-3x3-64 (groupsort 2) |
| conv-3x3-64 (groupsort 2) |
| invertible downsampling |
| conv-3x3-128 (groupsort 2) |
| conv-3x3-128 (groupsort 2) |
| flatten |
| dense-128 (groupsort 2) |
| dense-101 (None) |

(a) Network architecture used in the consistency experiment. It has 1.6M trainable parameters.

| parameter | value |
|---|---|
| data augmentation | None |
| input scale | $[0, 1]$ |
| batch size | 1000 |
| learning rate | 1e-5 |
| optimizer | Adam |
| cosine decay | None |
| epochs | 300 |

(b) Training parameters used in the consistency experiment. No data augmentation has been used, as it would artificially increase the number of samples in the dataset and biases the results.

Figure 9

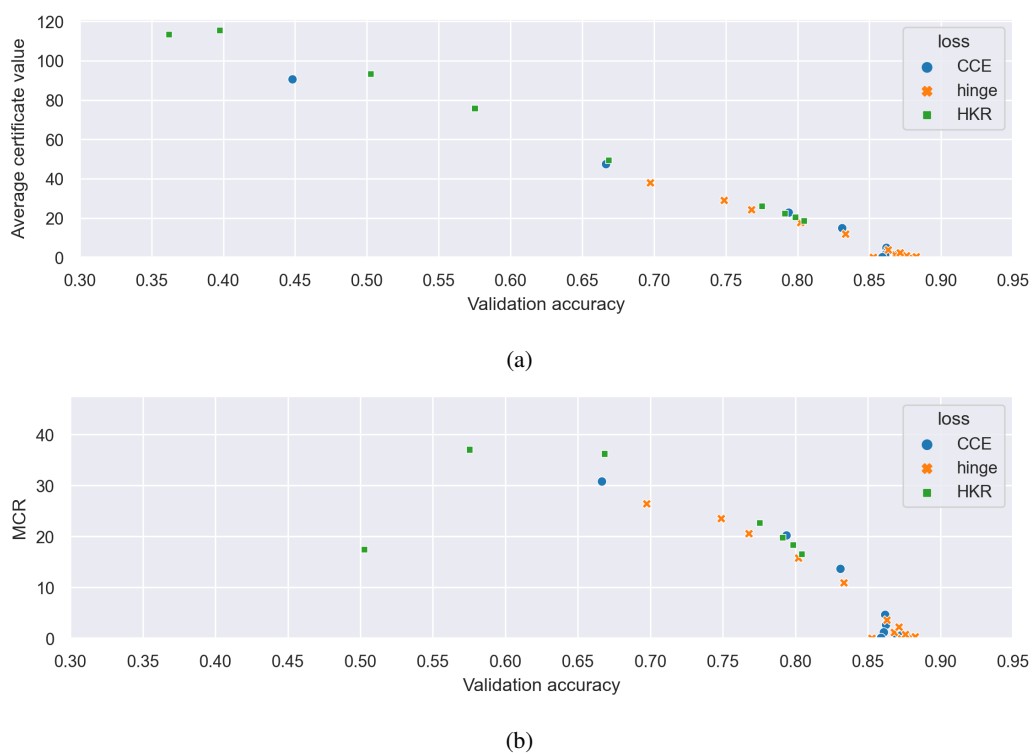

(a)

(b)

Figure 10: Pareto front for other robustness metrics: depending on the metric chosen to evaluate robustness, the shape of the Pareto front is changed. Upper chart shows use the average certificate value (robustness that does **not** take into account the true label, only the average value of $|f(x)|$), while the lower uses the MCR. The same models are used for these two graphs and Fig 3.

## K Controlling the accuracy/robustness tradeoff (experimental protocol of figure 3)

A small CNN architecture, described more precisely in Figure 11a was trained multiple times with different losses and different loss parameters. Besides the learning rate, other parameters were left unchanged, and are depicted in Figure 11b. The learning rate is chosen depending on the loss parameters: when trained with CCE, changing $\tau$ implicitly changes the norm of the gradient, thus when doubling $\tau$ one must divide the learning rate by a factor of two. The same phenomenon occurs with the $\alpha$ parameter of HKR. We kept $m = 20$ fixed for HKR, and we tuned $\alpha$.

For each run the final validation accuracy is reported on **x**-axis while the provable accuracy at $\epsilon = 36$ is reported on **y** axis. The choice of the robustness metric was set to the robust accuracy at $\epsilon = 36$ because of its wide use in the community. However, other robustness metrics also yield a Pareto front: two examples are shown in Figures 10a and 10b. In those examples, we also test different combinations of values for $m$ in HKR.

Note that this Pareto front can also be influenced by other factors: training larger architectures could improve both accuracy and robustness. Similarly, data augmentation has an impact on accuracy and robustness, but studying the phenomenon is out of the scope of this paper.

Finally, it is important to note that, the comparison between two architectures (or two robustness methods) cannot be done properly with a single training (and fixed hyper-parameters): comparing their Pareto front is more relevant.

| parameter | value |
|---|---|
| data augmentation | |

- random flip left right
- random brightness: $\delta = 0.2$
- random contrast: lower=0.75, upper=1.3
- random hue: $\delta = 0.1$
- random saturation: lower=0.8 upper=1.2
- random crop: scale $\in [0.8, 1.0]$

| | |
|---|---|
| input scale | $[0, 256]$ |
| batch size | 512 |
| learning rate | $[5 \times 10^{-2}, 1 \times 10^{-2}, 5 \times 10^{-3}, 1 \times 10^{-3}]$ |
| optimizer | Adam |
| cosine decay | 1e-2 |
| epochs | 300 |

| network architecture |
|---|
| conv-3x3-32 (groupsort 2) |
| conv-3x3-32 (groupsort 2) |
| L2 norm pooling 2D |
| conv-3x3-64 (groupsort 2) |
| conv-3x3-64 (groupsort 2) |
| conv-3x3-64 (groupsort 2) |
| L2 norm pooling 2D |
| conv-3x3-128 (groupsort 2) |
| conv-3x3-128 (groupsort 2) |
| conv-3x3-128 (groupsort 2) |
| global L2 norm pooling 2D |
| dense-128 (groupsort 2) |
| dense-101 (None) |

(a) Network architecture used in the Pareto front experiment. It has 0.4M trainable parameters.

(b) Training parameters used to build the Pareto front between accuracy and robustness. As the loss parameters implicitly change gradient norm, learning rate has been changed adequately, ranging from $5e - 2$ (low $\tau$ and low $\alpha$) to $1e - 3$ (high $\tau$ and high $\alpha$).

Figure 11

## L    Hardware

Toy experiments depicted in Fig. 1b, Fig. 1a, Fig. 2 and example 1 were run on a personal workstation with NVIDIA Geforce 1080 GTX and 8GB VRAM, 16 cores Xeon and 32GB RAM.

Large scales experiments depicted in Fig. 4, Fig.3 were run on Google Cloud with TPU v2-8. For reference, the experiments with CNNs on CIFAR10 (appendix K), took 4.9s per epoch on average.

*Tensorflow* framework was used in every experiment but the one of Example 1, where *Jax* was used instead (because order2 and *float64* experiments are easier to write in this library).

## M    Divergence of the weights on AllNet networks

In this example, we illustrate that example1 behavior can be observed at larger scale on MNIST with a ConvNet of AllNet . We used $3 \times 3$ convolution filters of widths $32 \rightarrow 64$ with **MaxPool** and **ReLU**, followed by a flattening operation and densely connected layers of widths $256 \rightarrow 10$.

Newton's method cannot be used due to its memory requirements on ConvNet. We tested SGD with learning rate $\eta = 0.1$ and momentum $m = 0.9$, and Adam with learning rate $\eta = 1e - 3$ and other default parameters. Experiments were run both in *float32* and *float64* precision. We monitor the maximum spectral norm of the weights of the network throughout training for each epoch $t \in \mathbb{N}$:

$$\mathcal{M}^t = \max_i \|W_i^t\|_2.$$

We report $\mathcal{M}^t$ as function of epoch $t$ in Figure 12. The validation accuracy is above 98% after the first epoch, and fluctuates between 98.5% and 99.5% during the following epochs (in either cases). Similarly the validation loss fluctuates between $1e - 1$ and $1e - 3$. We see that on this simple task the spectral norm of weight matrices continues to grow indefinitely, even though the classifier is almost perfect after one epoch. Interestingly, on this experiment the vanishing gradient phenomenon cannot be observed after 25 epochs and the results are robust with respect to the precision of the floating point arithmetic.

This is compliant with the observations made in the literature about the high Lipschitz constant of AllNet networks [23]. We observe that Adam makes the problem worse, even if its learning rate is smaller. This may explain why many practitioners reported that Adam was more susceptible to overfit than SGD with a carefully tuned learning rate scheduling.

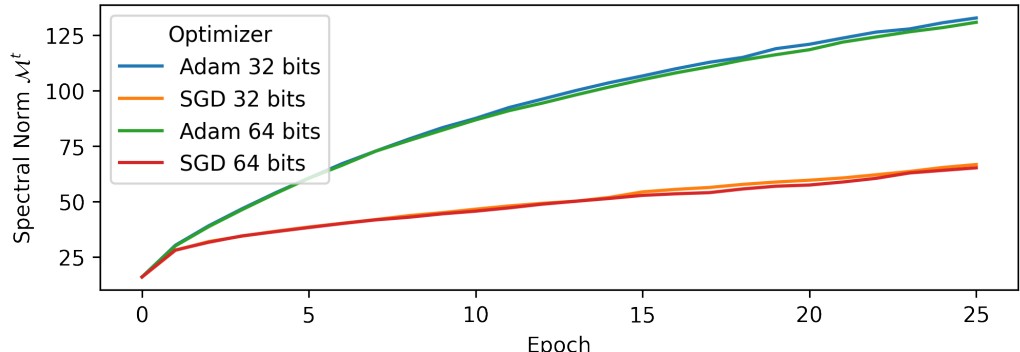

Figure 12: Maximum spectral norm of the weights of a simple ConvNet of AllNet trained with different optimizers on MNIST dataset. The validation accuracy remains above $98.5\%$ after the second epoch but the network's weights do not converge: the spectral norm seems to grow indefinitely.

# N  Stability of training of LipNet1

To check this, we trained different LipNet1 networks, either by tuning the value of temperature $\tau$, or by tuning the number of filters in convolutional layers. We used Fashion-Mnist dataset.

## N.1  Moving along Pareto front by tuning temperature

In this experiment, we explore the stability of training with respect to the loss parameter (here we use CCE with $\tau$). To do so we perform a scheduling on $\tau$: each network of the experiment is trained with a fixed $\tau$ for 200 epochs, then $\tau$ is increased/decreased linearly by a factor of 2 for 200 more epochs (see 13. We train a total of 12 LipNet1 networks with the same architecture: three blocks of two convolutions, bias and group sort are followed by a Pooling layer (L2NormPooling) finally followed by a flatten and

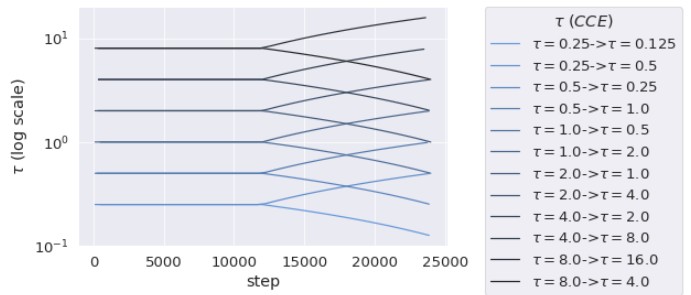

Figure 13: Schedule for the $\tau$ parameter: $\tau$ is set to be constant for 200 epochs, followed by a linear increased/decreased by a factor of 2 for 200 epochs.

a Dense layer (architecture synthesized as c32-c32-P-c64-c64-P-c128-c128-D256 ). Each training is performed with the same optimizer (Adam) and the same learning rate (0.001). Each dot in the graph of Figure 16 corresponds to the metrics of a network after one epoch. The validation accuracy can be found on **x-axis** and MCR metric on **y-axis**.

During the first epochs, the dots can be found inside the region delimited by the Pareto front: the network has not converged yet, and both Mean Certifiable Robustness and validation accuracy are low. After few epochs, the dots start to accumulate on the Pareto front. Then, the value of temperature $\tau$ is tuned *during* the training, from initial $\tau_{\text{start}}$ to $\tau_{\text{final}}$. Each of the color corresponds to a different value of Tau. We see that the temperature can be modified during training to move along the Pareto front.

We can get a closer look at the trajectory of two networks, which are reported in fig 15 to better illustrate this phenomenon. Despite their starting point being different, they end up on a minimum with the same MCR/accuracy tradeoff. It seems that the position on the Pareto front only depends on the value of $\tau_{\text{final}}$. Not only the functions are similar at a global scale, but it is also valid at a local scale: while the two nets are only 77% accurate, they agree on 93% of the validation samples (ie. they make the same error on the same sample).

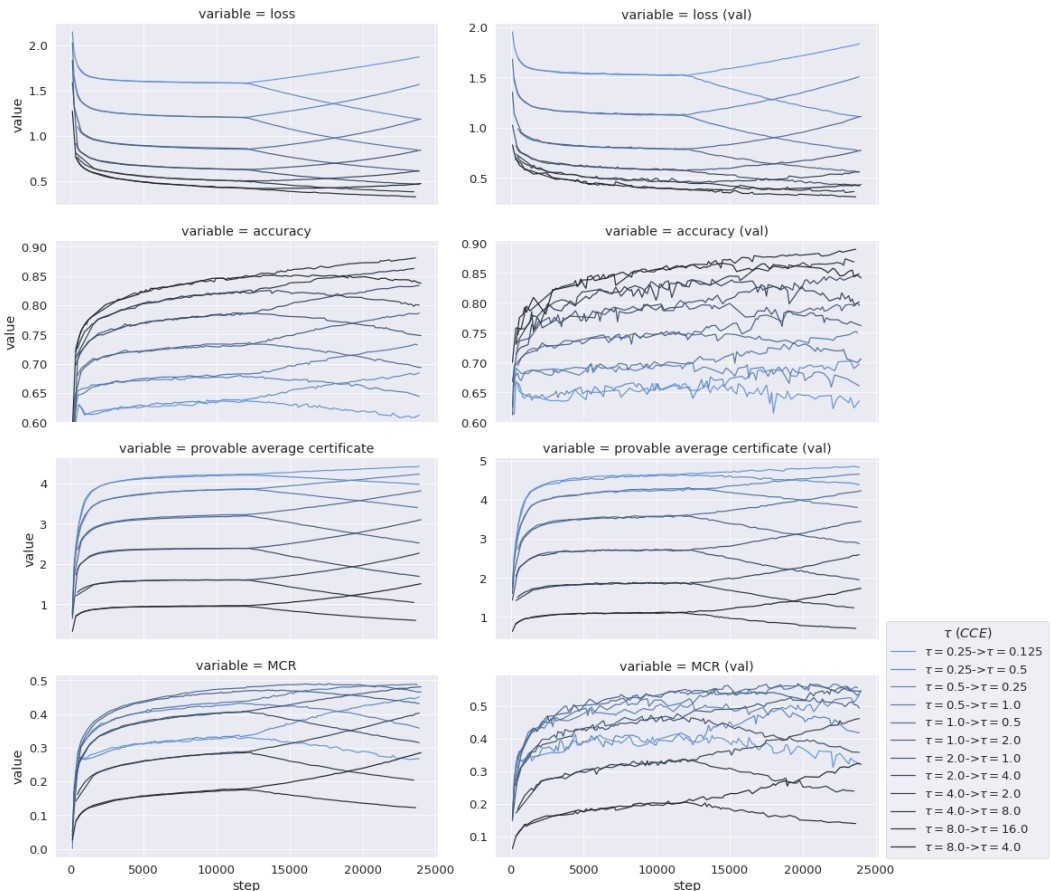

Figure 14: **Learning curves on Fashion-Mnist**: 12 LipNet1 networks are trained using the schedule depicted in fig 13 with respectively $\tau_{\text{init}} \in [0.25, 0.5, 1, 2, 4, 8]$. At each epoch accuracy (x axis) and MCR (y axis) are reported.

Observe that all dots tend to accumulate on the Pareto front, even though they are 12 different networks being trained. It suggests that this method is stable with respect to the input seed. Some of the networks trained with high $\tau_{\text{start}}$ (for example $\tau_{\text{start}} = 8$ and $\tau_{\text{final}} = 16$) seem to "lag behind": empirically we observe that more epochs are required to make the network converge. Hence, the speed at which $\tau$ is modified must be scaled appropriately to ensure that the best Pareto front is recovered.

This experiment also suggests that a curriculum can be a satisfying approach to tune $\tau$ : an expensive grid search over $\tau$ could be replaced by a single training with a scheduler placed on $\tau$.

| model | model1 | model2 |
|---|---|---|
| $\tau_i$ | 1.0 | 4.0 |
| $\tau_f$ | 2.0 | 2.0 |
| accuracy (train) | 0.7838 | 0.7793 |
| coincidence (train) | 0.9421 | |
| accuracy (val) | 0.7754 | 0.7724 |
| coincidence (val) | 0.9350 | |
| coincidence (random) | 0.9937 | |

Figure 15: two LipNet1 networks with different initialization and learning curriculum learns similar function as long as the $\tau$ is the same at the end. Although these models only have 69% accuracy, their predictions match on 92% of the test set samples.

### N.2 Shifting Pareto front by tuning architecture

In this experiment we explore an important question: what happens when the architecture is changed? To explore this we perform the same experiment as N, but with smaller and larger architectures: the

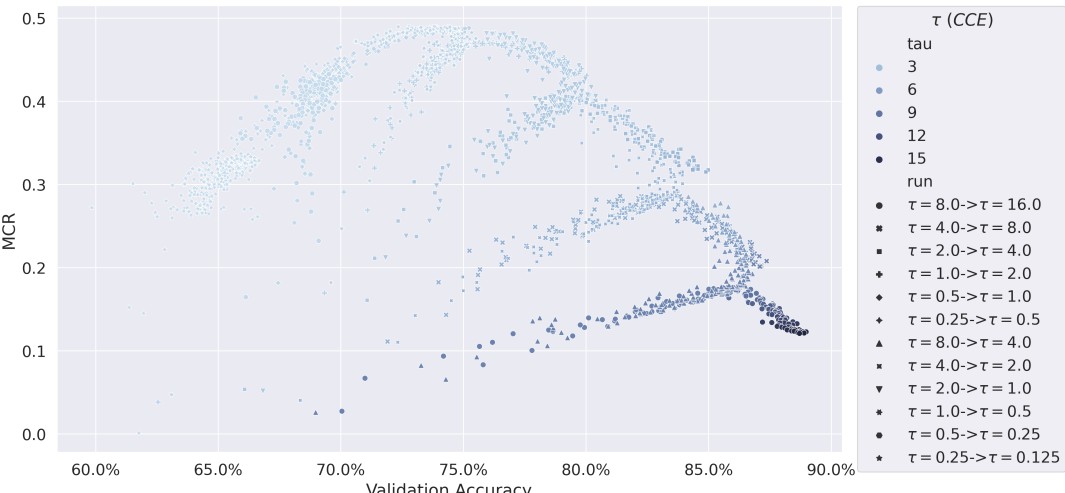

Figure 16: **MCR/accuracy tradeoff on the validation set of Fashion-Mnist**: 12 LipNet1 networks are trained using the schedule depicted in fig 13 with respectively $\tau_{\text{init}} \in [0.25, 0.5, 1, 2, 4, 8]$. At each epoch accuracy (x axis) and MCR (y axis) are reported. The Pareto front is still apparent: the MCR/accuracy tradeoff only depends on $\tau$ and not on the initialization.

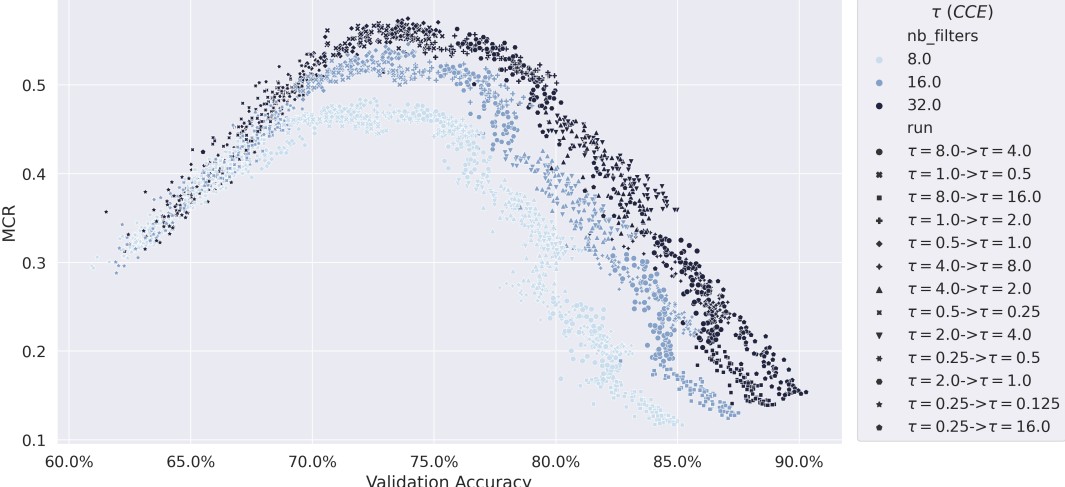

Figure 17: **MCR/accuracy tradeoff on Fashion-Mnist for LipNet1 when architecture size changes:** At each epoch, validation accuracy and MCR are reported. The networks are trained following a scheduling for $\tau$ as described in Figure 13. We see that larger networks are more expressive, and the Pareto front is shifted toward higher accuracy and higher robustness.

architectures are denoted by the number of filters in their first convolutions (filter of other block are adjusted accordingly by doubling the number of filter of the previous block). It shows how the expressiveness of the architectures affects the Pareto front. We report the results in Figure 17.

The validation accuracy can be found on **x-axis** and the MCR on **y-axis**. Each dot corresponds to an epoch/a network. Different colors correspond to different architecture widths.

As expected, larger networks are more expressive, and as a result, the Pareto front is shifted toward higher accuracy and higher robustness. This observation holds for every scheduling $\tau$. The MCR/accuracy is also architecture dependent.