# OpenReview forum: "Pay attention to your loss : understanding misconceptions about Lipschitz neural networks"
_NeurIPS.cc/2022/Conference — NeurIPS 2022 Accept_

### Official Review · Reviewer_VK8y · 2022-07-10

**Rating:** 6
**Confidence:** 3
**Soundness:** 3 good
**Presentation:** 3 good
**Contribution:** 3 good

**Summary:**

This paper challenges the common belief that 1-Lipschitz natural networks are less generalizable than regular networks. It accomplishes this by proving a series of claims. The authors first demonstrate that any classification problem, under mild assumptions, has a 1-Lipschitz solution, and that the error is zero if the distributions of classes are well separated. They go on to show that 1-Lipschitz models can always minimize a tempered cross-entropy loss, while the generalization performance depends heavily on the temperature value. This paper suggests that the low performance of 1-Lipschitz networks in the literature can be attributed to the wrong choice of temperature. One positive thing about 1-Lipschitz networks is that one can certify the output margin $\|f(x)\|$ rather than the input margin, as the former is a lower bound of the latter. The paper demonstrates that the WGAN discriminator maximizes an average of the signed output margin, thus it is optimally robust in this respect. The WGAN discriminator has the disadvantage of not ensuring a very high level of accuracy. There are even classification problems for which the WGAN discriminator's accuracy can be arbitrarily low. Nevertheless, implicit/explicit regularizations with 1-Wasserstein distance can be used for a trade-off between accuracy and “certifiable” robustness in 1-Lipschitz networks. Finally, the paper discusses some guarantees on the learnability of 1-Lipschitz networks.

**Questions:**

- Isn’t the requirement in Eq. (3) too conservative? Does changing the min over x to an expectation help here?
- Are you sure about Lines 182 and 183? The MCR for Q is not missing here?
- Can you provide an example of the behavior discussed at the end of Sec. 5.2 for conv nets trained on datasets like CIFAR-10 or MNIST?
- There is no comparison with SotA certifiable networks in the current manuscript.
- Adversarial training of regular networks leads to a large gap between training and test losses. Should we expect this gap to disappear for adversarial training of 1-Lipschitz networks?
- Do the authors think that playing with the temperature of 1-Lipschitz networks is enough to build certifiable robust classifiers?
- The observation made in this paper bears some similarities to other publications, such as (On Calibration of Modern Neural Networks, Guo et al., ICML 2017) and (How Good is the Bayes Posterior in Deep Neural Networks Really?, Wenzel et al., ICML 2020).
- Some typos: L137, cannot guarantees; L138, “is not” repeated twice; L154: an unified.

**Limitations:**

The extent to which certain observations of this paper hold in practice is not clear enough.

**Strengths And Weaknesses:**

This is a thought-provoking paper that questions certain overlooked aspects of 1-Lipschitz networks that are associated with adversarial robustness. It contains many interesting insights as well as simple illustrations to demystify some myths about these networks. While discussing interesting aspects of Lipschitz networks, the paper does not present its results in a unified manner, rather it provides nuggets of understanding. I am pleased to learn that 1-Lipschitz networks may not be as weak as previously thought, but I would like to know how these insights can be used to build robust and accurate models. Additionally, the paper could have better positioned itself compared to previous studies.

---

> ### Author Response · Authors · 2022-08-01
> **Answer to questions**
>
> Thank you for your detailed comments and questions.
>
> >  There is no comparison with SotA certifiable networks in the current manuscript.
>
> Please see our general response regarding this remark.
>
> > Isn’t the requirement in Eq. (3) too conservative? Does changing the min over x to an expectation help here?
>
> It is too conservative for practical purposes: with the “min” we enjoy the nice geometric interpretation (Signed Distance Function), whereas the “expectation” is a lot harder to analyze.
>
> > Can you provide an example of the behavior discussed at the end of Sec. 5.2 for conv nets trained on datasets like CIFAR-10 or MNIST?
>
> We added an experiment that confirmed our remark: see please our general response.
>
> > Adversarial training of regular networks leads to a large gap between training and test losses. Should we expect this gap to disappear for adversarial training of 1-Lipschitz networks?
>
> Thanks for this question, as you pointed out adversarial training can be combined with LipNet1 networks. However, as this method attempts to do empirically what LipNet1 does with usual training, combining the two might be redundant. When the loss parameters are not tuned properly LipNet1 networks can also have a large generalization gap, as shown in fig 4.
>
> > Do the authors think that playing with the temperature of 1-Lipschitz networks is enough to build certifiable robust classifiers?
>
> It is necessary, but not sufficient. The robustness certificate exists by design (Property 1). The temperature will define the nature of the optimum (see figures 1.b, 2). For the certificate to be tight, the upper bound on Lipschitz constant must be tight. Hence, the architecture plays an important role in the final quality of the certificates, as does the overall optimization process. For instance, designing orthogonal convolutions is still an active research area.
>
> > The observation made in this paper bears some similarities to other publications, such as (On Calibration of Modern Neural Networks, Guo et al., ICML 2017) and (How Good is the Bayes Posterior in Deep Neural Networks Really?, Wenzel et al., ICML 2020).
>
> Thanks for the additional references ! Note that the LipNet1 network is not necessarily calibrated : the value of the probability sigma(f(x)) is more an indication of the distance to the boundary (a lower bound at least), than a confidence like the one encountered in calibration.
>
> > The extent to which certain observations of this paper hold in practice is not clear enough
>
> We hope that experiments we added on Fashion-Mnist with different architectures (appendix N), in addition to the ones already present on CIFAR-10 and CIFAR-100, can convince you the phenomenon we exhibit holds broadly. We also added an experiment on Mnist to show that Example 1 continues to hold.
>
> > Are you sure about Lines 182 and 183; Some typos: L137, cannot guarantees; L138, “is not” repeated twice; L154: an unified.
>
> We have corrected these typos, thanks for your careful reading.

---

> > ### Comment · Reviewer_VK8y · 2022-08-09
> > **Thanks for answering my questions**
> >
> > Honestly, I could not read the whole paper again but I trust my fellow reviewers that its coherence has improved. I am hopeful to see this work accepted at NeurIPS 2022.

---

> > > ### Author Response · Authors · 2022-08-09
> > > **[Re] Thanks for answering my questions**
> > >
> > > Thank you for your kind words.

---

### Official Review · Reviewer_eWSm · 2022-07-11

**Rating:** 3
**Confidence:** 4
**Soundness:** 3 good
**Presentation:** 1 poor
**Contribution:** 2 fair

**Summary:**

The paper proposes many theoretical statement Lipschitz neural network for classification and show that these specific networks follows attractive properties.

**Questions:**

I suggest that the other should carefully rewrite the paper with a clear objective, the typography should be respected.
The different results of the paper are nice, but in order to be published they should be properly articulated.

**Limitations:**

Yes

**Strengths And Weaknesses:**

AllNet networks are defined as any neural networks without any constraints. This class does not produce Lipschitz function without any further hypothesis (such as activations are Lipschitz).

The paper is all over the place and the main research line is unclear. Most of the theoretical results are rather common or "trivial". The whole paper seeks for internal coherence and lacks of general purpose. As a reader, it is difficult to grasp what is the objective.
The present paper seems to be more adequate for a journal publication than to a conference.

* Global style as the biggest weakness

The main assumption behind the development of the theory is the fact that Lipchitz neural networks are "often perceived as not expressive", which I do not remember ever reading in the literature. Most of the paper seems fuzzy: it goes from BCE to robustness certification to a (trivial) convergence proposition (prop 4) to consideration about float32/64 (ex 1)... it is just too much. There are no transitions and at no moment the reader understands what is the point.

One of the biggest problem is the writing style that makes the paper hard and annoying (sorry...) to read.
For example among many other problems:

    - sentences must contain a verb and end with a '.'(even when it ends with a formula);
    - theorem should be self contained, exterior references should remain exceptional;
    - organize the equations in order to be easily readable;
    - separate definitions and remarks (e.g. def 1 & 2 contains definitions and remarks);
    - separate propositions and definitions (e.g. corollary 1 contains a definition and a proposition);
    - the proofs rely on many technical tools such as Lebesgue measure but the pre-requisite (Borel space etc) are never explicitely given.
Most of the proofs in the appendix should be re-written. More personal but I think the footnotes in the paper could be avoided.
The paper is full of references and it costs the readability of the arguments. The reader gets overwhelmed very quickly for the wrong reasons.

* References

most reference are incomplete (many article are reference as arxiv instead than the published and peer-reviewed version)

* Others

l94: one needs additional hypothesis on AllNet to claim that they have finite Lipschitz constant (Lipschitz activation would do).

l216: the reference to cor 1 seems dubious: it is a definition of \epsilon separation rather than about bias.

---

> ### Author Response · Authors · 2022-08-01
> **Answer to concerns**
>
> Thank you for your comment regarding typography rules. We do appreciate that you found our results nice, we are also convinced that such a paper is important for the NeurIPS community to increase interest on LipNet1 networks.
>
> Concerning the general writing style of this paper, we invite you to read the general response.
>
> > the biggest problem is the writing style
>
> We modified definitions 1&2 (we added a remark), we added a definition before corollary 1. We rewrote some equations. We moved the footnotes into the main flow. We are confident that the remaining remarks regarding typography can be easily resolved in the respect of the 9 pages limit, upon acceptance.
>
> Concerning the proof readability, we are prone to rework on the proofs. Can you elaborate on the style you would like to read?
>
> > - the proofs rely on many technical tools such as Lebesgue measure but the pre-requisite (Borel space etc) are never explicitly given.
>
> Thanks for your remark. We will add definition /or references in the appendix in order to make the paper self consistent with regards to these classical concepts of measure theory.
>
> > Lipchitz neural networks are "often perceived as not expressive", which I do not remember ever reading in the literature
>
> The citation given at the end of the first page “Lipschitz-based approaches suffer from some representational limitations that may prevent them from achieving higher levels of performance and being applicable to more complicated problems” comes from a peer reviewed paper [9].
>
> > The paper is full of references and it costs the readability of the arguments
>
> This is something we discussed during the writing. We wanted to support every claim by citing appropriately prior works. We are advocating for the appealing properties of those networks: it requires a lot of tools and we cannot afford to put them all in the main paper. We had to choose the most significant parts to leave space for the main message, and give pointers to allow the reader to navigate the remainder in references or appendix.
>
> > most reference are incomplete (many article are reference as arxiv instead than the published and peer-reviewed version)
>
> Indeed 12 out of the 84 citations were referenced as arxiv. Among them for 5 papers [32, 38, 54, 68, 73] we couldn’t find a peer reviewed equivalent on the authors webpages. For the others we amended the manuscript, including the ones published this year like [44, 48, 67].
>
> > l94: one needs additional hypothesis on AllNet to claim that they have finite Lipschitz constant (Lipschitz activation would do).
>
> We amended the manuscript. The vast majority of activations have a finite lipschitz constant (ReLu, Softplus, Sigmoid, Tanh, GeLU, etc). All details related to this result can be found in the paper [16] we are referring to.
>
> > l216: the reference to cor 1 seems dubious: it is a definition of \epsilon separation rather than about bias.
>
> We are referring to the bias/variance tradeoff in learning. Corollary 1 is saying that under the \epsilon separation hypothesis, 100% accuracy is achievable by LipNet1 networks. It implies that the LipNet1 class does not suffer from any bias. We rewrote the Corollary to make it more clear.

---

> > ### Comment · Reviewer_eWSm · 2022-08-08
> > **Thank you for updating**
> >
> > Thank you very much for your answer and sorry for the delay. I read the updated version and it is indeed style-wise much much better. I would improve my grade following this improvement, but in a limited way: despite your clear answer about the objective of your paper, I still fail to find a definite general purpose to your work that I still see as a collection of unsurprising (it is not a bad thing!) facts about Lipschitz Deep Neural Networks (that are kind of all network as stated from Definition 1).
> > An interesting potential direction in my opinion would be to shed some light on which Lispchitz constraint technic should be used for which problem, and can we distinguish Lipschitz network from one another. Could you work tackle this kind of problem?

---

> > > ### Author Response · Authors · 2022-08-09
> > > **Thank you, and answer to your question**
> > >
> > > Thank you for your thoughtful answer and your willingness to improve your rating. We have worked toward the changes you suggested. We updated the appendices to add definitions of the tools used.
> > >
> > > > a collection of unsurprising (it is not a bad thing!) facts about Lipschitz Deep Neural Networks
> > >
> > > Our experience on the matter is that it depends on the mathematical culture of the reader. Some researchers are surprised by these results. To the best of our knowledge, the link between cross-entropy temperature and accuracy/robustness tradeoff have not been studied before. The ill-posed problem of BCE minimization on AllNet is often not addressed or mentioned. We would not qualify those facts as unsurprising.
> > >
> > > > that are kind of all network as stated from Definition 1
> > >
> > > It's true that 1-Lipschitz networks are a subset of conventional networks. The key difference is in what the optimization process yields. Crucially, our work shows that
> > >
> > > $$\bar f = \arg\min_{f\in\text{LipNet1}} \mathbb{E}_{(x,y)\sim P} [\mathcal{L}(f(x),y)] \text{ (exists under mild assumptions per Prop 2)}$$
> > >
> > > is very different from
> > >
> > > $$\begin{aligned}f &= \arg\min_{f\in\text{AllNet}} \mathbb{E}_{(x,y)\sim P} [\mathcal{L}(f(x), y)] \text{ (assuming it exists, not always per Prop 5)},\\\\
> > > \bar f &= \frac{f}{\text{Lip}(f)}\end{aligned}$$
> > >
> > > both theoretically and empirically.
> > >
> > > > which Lispchitz constraint technic should be used for which problem, and can we distinguish Lipschitz network from one another.
> > >
> > > Thank you for opening the discussion on possible extensions of our work. This is a good remark that highlights the importance of exploring this field, by keeping in mind both the role of architecture and the role of the loss.
> > >
> > > First, let's recall some facts from literature.
> > >
> > > * ReLU based networks suffer from expressiveness issues (pointed out by [11]) and should be replaced by GroupSor (known to be superior [36]).
> > > * The quality of robustness certificate depends on the tightness of the upper bound on Lipschitz constant. So methods based on regularization (such as spectral regularization [38] or gradient penalty [37]) should be avoided since they do not allow precise control of the Lipschitz constant.
> > > * Frobenius normalization or weight clipping is better [6, 39] but it often yields loose upper bounds wrt l2 distance. Spectral normalization [14] leads to tighter bounds.
> > >
> > > We can share a preliminary answer in the context of classification.
> > >
> > > * To ensure tightness of robustness certificate, the robustness radius at $x$ must fulfill $\epsilon = |f(x)|$ which implies $\|\nabla_x f(x)\|=1$. It is possible per Prop 1 and Corollary 2. This property can be enforced with orthogonal layers (including GroupSort), and orthogonal affine layers (in particular their spectral norm is equal to 1). Or by using hKR loss whose solution verifies this property by design [8] without explicit orthogonal constraints.
> > > * hKR and hinge yield VC bounds per Prop 6 because they separate samples based on margins. Ubiquitous in C-SVM, hinge loss and its variants are not frequently used for NN training. Our work may help rehabilitate them: the margin must be tuned (not possible with default implementation in Tensorflow framework).
> > > * The role of $\tau$ in cross-entropy is to control the accuracy-robustness tradeoff (illustrated by fig 2) - and sensitivity to noise (as in fig 1.b).
> > > * Please take a look at the discussion with reviewer jRZ5 for additional insights regarding other architectures.
> > >
> > > We plan on exploring those questions in future work. We will add a “Perspectives” section that contains this discussion in the additional page of the camera ready version.

---

> > > > ### Comment · Reviewer_eWSm · 2022-08-09
> > > > **answer to answer**
> > > >
> > > > Thank you for your answer and the proposed developements. I will need some time to fully process this discussion, thank you very much for the interaction!

---

### Official Review · Reviewer_wa64 · 2022-07-12

**Rating:** 5
**Confidence:** 3
**Soundness:** 2 fair
**Presentation:** 2 fair
**Contribution:** 3 good

**Summary:**

The authors theoretically studied several properties of 1-Lipschitz neural networks (LipNet1). They showed that LipNet1 can learn and approximate any classification decision boundary. Secondly, they showed that LipNet1 is certifiable robust and not prone to overfitting, unlike conventional unconstrained models. They showed that by varying margin for hinge loss or temperature for binary cross-entropy, one can control the trade-off between robustness and accuracy.

**Questions:**

- Can the authors include experiments where LipNet1 model robustness with tuned $L$ or $\tau$ is tuned and compared with current state-of-the-art methods?
- How does the choice of the loss function affect the training dynamics and generalization?

**Limitations:**

The authors briefly discussed the limitations of their work: lack of convergence speed bounds. The limiations of this work can be expanded. The authors should describe all the assumptions in detail and clearly state their limitations when providing their theoretical results.

**Strengths And Weaknesses:**

**Originality:**

- The paper derives several theoretical results. Yet, I am not confident to say that all the results presented here are original.

**Quality:**

- The authors state that the purpose of this paper is to address the common misconception that “they (1-Lipschitz neural networks) remain commonly considered as less accurate”. However, the authors do not cite any experimental study or theoretical work, where someone claimed that 1-Lipschitz neural networks are inherently less accurate or less expressive than unconstrained neural network models.
- The authors provide an interesting analysis of 1-Lipschitz neural networks and show that LipNet1 can have certain advantages over conventional neural networks.
- The theoretical analysis, however, is not very useful as 1) we do not know how many samples are required for the convergence of the LipNet1 models; 2) the initial VC dimension bounds are rather vacuous, see proposition 7, where VC bounds depend exponentially on the number of neurons.

**Clarity:**

- The paper at times is difficult to follow its main story is fragmented without sharing a common theme. The expanded experiments (not just toy experiments) should be included to expand the paper and confirm the main theoretical results.

**Significance:**

- The paper provides an analysis of the properties of LipNet1 models, which recently become of interest in the community recently. However, to highlight its significance, the authors should expand the experimental section to compare LipNet1 models and conventional models. At the present moment, the provided evidence that LipNet1 models are not prone to overfitting unlike conventional models is not conclusive.

---

> ### Author Response · Authors · 2022-08-01
> **Answer to question and concerns**
>
> Thank you for your careful reading and detailed questions.
>
> > the authors do not cite any experimental study or theoretical work, where someone claimed that 1-Lipschitz neural networks are inherently less accurate or less expressive than unconstrained neural network models.
>
> We gave a citation line 34 from paper [9]:  “Lipschitz-based approaches suffer from some representational limitations that may prevent them from achieving higher levels of performance and being applicable to more complicated problems”.
>
> > the provided evidence that LipNet1 models are not prone to overfitting unlike conventional models is not conclusive
>
> Benefitting from generalization guarantees does not exactly imply that lipschitz networks are not prone to overfitting. Proposition 4 proves that once the loss parameters are fixed, the generalization gap vanishes to zero *in the limit* of large sample size. Figure 4 shows that this phenomenon can be observed in practice. For the practitioner, this means that generalization gap can be reduced by:
> * increasing the number of samples once tau is fixed (or other loss parameter).
> * decreasing tau once the number of samples is fixed
>
> We added curves of AllNet networks in Fig 4. to illustrate their generalization gap.
>
> > the authors should expand the experimental section to compare LipNet1 models and conventional models
>
> We added an experiment with an AllNet network in Section 5.2 (fig 12) and in fig 4. Please see our general answer.
>
> > Can the authors include experiments where LipNet1 model robustness with tuned L or tau is tuned and compared with current state-of-the-art methods?
>
> Please see our general response regarding this remark.
>
> > How does the choice of the loss function affect the training dynamics and generalization?
>
> This is an interesting question. Our results (figure 3) reveal that the studied losses share a similar behavior at global scale at least from the point of view of accuracy, robustness and generalization. Whether these losses differ at a local scale is still an open question. We give insights about this in fig 14.
>
> The remarkable stability of LipNet1 training can also be observed in the additional experiment on fashion-mnist (fig 15).
> Differences regarding other criterions (such as explainability) might be expected. hKR enjoys a nice optimal transport interpretation [8]. However, hKR and hinge are piecewise affine (like LipNet1 networks), so they cannot be trained with order-2 methods.
>
> > We do not know how many samples are required for the convergence of the LipNet1 models:
>
> fig 4 shows that this convergence can be observed empirically (see fig 8. for detailed results). We believe we can obtain convergence rates using Donsker’s theorem, since Lipschitz functions are a Donsker’s class [90] under mild assumptions [91], but we leave this for future work.
>
> [90] Aad W. Van Der Vaart et Jon. A. Wellner, Weak convergence and empirical processes with applications to statistics, Springer, p. 127
>
> [91] Giné, E. and Zinn, J., 1986. Empirical processes indexed by Lipschitz functions. The Annals of Probability, pp.1329-1338.
>
> We add details in Section 5.1 about this limitation of our work.
>
> > The initial VC dimension bounds are rather vacuous:
>
> We emphasize that existing proofs that assume elementwise activation, are not applicable to GroupSort which is widely used in LipNet1. As expressed in l267, we are pretty confident that future works could improve this bound.

---

### Official Review · Reviewer_jRZ5 · 2022-07-14

**Rating:** 7
**Confidence:** 2
**Soundness:** 4 excellent
**Presentation:** 2 fair
**Contribution:** 4 excellent

**Summary:**

The paper studies 1-Lipschitz constrained neural networks, and prove some results for this class of networks. The paper defines 1Lip nets as those networks for which the output function is Lipschitz constrained with a Lipschitz constant of 1. They cite a way to construct such feed forward networks based on the norms of the weight matrices. They then show that for any binary labelling function, one can get an equivalent 1-Lipschitz function which agrees with the labelling function on all points of the domain. They then empirically show that tuning the temperature hyper-parameter while learning with BCE loss is important, and imply connections between this hyper-parameter and the lipschitz constant. They then show that 1Lip network are certifiably robust to adversarial attacks, and that the discriminator of a WGAN, which is also a 1Lip network is the optimally robust classifier. Finally, they give generalization results of 1Lip networks, showing that this class of networks is PAC learnable with a VC dimension independent of the architecture.

**Questions:**

How does this analysis extend to other types of networks apart from feed forward networks?

**Limitations:**

Some of the limitations are listed in the weaknesses above. A more thorough empirical evaluation on other tasks and adversarial perturbations would be appreciated.

**Strengths And Weaknesses:**

Strengths -
1. The results in the paper are novel, interesting and important.
2. The generalization guarantees and robustness results can open up a new line of research into these architectures.

Weaknesses -
1. More empirical evaluation could benefit the paper.
2. The writing could be improved, in particular, a lot of prior knowledge is assumed.

---

> ### Author Response · Authors · 2022-08-01
> **Answer to question**
>
> Thank you for your comprehensive reading, we answer your questions here, and we invite you to read the general response where the additional experiments are listed.
>
> > How does this analysis extend to other types of networks apart from feed forward networks?
>
> Thanks for your question which opens nice perspectives for future works.
> Effectively the focus of this paper was on feed forward networks which are the most common networks for computer vision (including convolutional layers). We have some preliminary results that we can share in this rebuttal.
>
> Self-attention layers are not Lipschitz (see [40]) so they do not fit in our analysis. Lipschitz Recurrent layers (such as LSTM) have been recently studied by [42]. Recurrent layers are useful for inputs that are given as sequences. Sequences of bounded length (<= l) can be embedded into euclidean space, eventually by padding sequences that are too short. If the latent space is big enough, the Lipschitz recurrent unit can carry the whole input and additional information, so we recover universal approximation. Hence, our work generalizes to Lipschitz recurrent units that take in input sequences of lengths <= l.
>
> However, for arbitrarily long sequences, we cannot embed them easily in euclidean space of finite dimension. Recurrent units can be used to simulate a Turing Machine [88] which might have huge implications on expressivity.
>
> In the case of Deep Equilibrium models [89] LipNet1 networks are contractive operators : the fixed point is guaranteed to exist by Banach fixed-point theorem. So they would be more sound (mathematically) with this implementation.
>
> [88] Hyötyniemi, H., 1996. Turing machines are recurrent neural networks. Proceedings of step, 96.
>
> [89] Bai, S., Kolter, J.Z. and Koltun, V., 2019. Deep equilibrium models. Advances in Neural Information Processing Systems, 32.
>
> > a lot of prior knowledge is assumed.
>
> We agree on that, especially about the practical implementation of LipNet1 networks, and mathematical tools. We hope that appendix D, E and F can help with practical implementation. We will add definitions in the appendix for the different tools we use such as VC dimension or optimal transport.

---

> > ### Comment · Reviewer_jRZ5 · 2022-08-04
> > **Response to the rebuttal**
> >
> > Thank you for your response. I would be interested in looking at the results for CNNs if you could share them.

---

> > > ### Author Response · Authors · 2022-08-05
> > > **CNN fall under the scope of the paper**
> > >
> > > Thanks for your interest. In fact CNN do fall under the scope of this paper, both on the theoretical and practical side: convolutions can be seen as a special case of a dense layer with sparse and repeated weights. But as you pointed out, CNN comes with it's own variety of building blocks: others layers must be analyzed independently: average pooling can be seen as a special case of convolution, skip connections must be followed by a 0.5 multiplicative factor, and so on... This (non-exhaustive) list motivated our choice for the deel-lip library (more information in appendix D), which allowed us to run large scale experiment with CNN on CIFAR-10/100, Mnist and Fashion-Mnist. We expect that our results still hold as this field evolves, so improvements can be expected on the experimental side (higher pareto front in fig 3, and convergence at an higher accuracy in fig 4 for instance). The importance of the CNN width is illustrated in fig 16 for example.

---

### Author Response · Authors · 2022-08-01
**General response (common concerns)**

We would like to thank the reviewers for their instructive feedback about our paper. From their review, it seems that the results of this paper are understood, including by reviewers who claimed to be “unfamiliar with some pieces of related works”. If we could convince you about the nice properties of LipNet1 for classification, we will have already reached our first goal.

## About the style of the paper
This paper aims to be at the intersection between theoretical ML and (empirical) deep learning. Lipschitz constrained networks allow to directly put in perspective mathematical proofs with empirical experiments on large scale vision datasets. The style is meant to be understandable by practitioners from the deep learning community, while staying mathematically rigorous. This motivated our choice to articulate each message with:
* theoretical result that carries the message of the section (Proposition 1, Proposition 2, Proposition 4)
* a toy experiment for pedagogical purposes (figures 1&2, example 1)
* a large scale experiment (figures 3, 4, 12, 15, 16)

Each message is carried by its own section, and summarized in Table 1. This organization is also reflected in the table of content given in the appendix, where each subsection title carries a message.

This structure allows us to show that, with LipNet1, theory matches the practice remarkably well (figures 2, 3 and 4), including on large scale experiments (fig 12, 15, 16). Some results (such as the consistency result of Proposition 4) have important consequences for the community of robust learning, and for practitioners interested in Deep Learning with formal guarantees (for safety critical applications in health or industry for example). Control of Lipschitz constant and training with LipNet1 extends further than the robustness community only, for example in fields like Differential Privacy [85], optimal transport [6], density estimation with orthogonal normalizing flows [58].

[85] Abadi, M., Chu, A., Goodfellow, I., McMahan, H.B., Mironov, I., Talwar, K. and Zhang, L., 2016, October. Deep learning with differential privacy. In Proceedings of the 2016 ACM SIGSAC conference on computer and communications security (pp. 308-318).

## About the research line of this paper
While the question of the LipNet1 architecture is often in the spotlight, the loss is overlooked. Our research line is to point out its tremendous importance (see sections 3.2, 4.2 and 4.3). In personal communications across different teams and labs, it became apparent that some teams assigned the poor results of their LipNet1 networks to an intrinsic limit of the architecture, without questioning their loss, and abandoned the tool before ever reaching production or publication stage. We feel that there is a gap in literature that must be filled.

This paper also provides a toolbox of results and experiments to serve as a basis for future works. We aim to open new research directions, and to encourage the community to use LipNet1 networks more, including outside the field of robust learning. We also believe it provides insight on what is going on in AllNet networks training: their Lipschitz constant grows uncontrollably during training (see fig 12), which is like using cross entropy with high Tau.

## About comparison with SOTA:
Comparing with state of the art methods in provable robustness has been discussed while redacting this paper. However doing so would have been counterproductive as:
1. All the networks in figure 3 (resp fig 4) share the same architecture, and training procedure. This allows a fair comparison. We explained in appendix D why we chose the Deel-Lip library, among other implementations of LipNet1 networks.
2. The conclusions of the paper does not depend on the exact architecture used to parametrize LipNet1 convolutions. By adding sota methods in the graphs, there is a risk of misleading the reader into thinking that we advocate a new method or a specific LipNet1 architecture.

In fact the results in fig 3 questions the way we evaluate the state of the art: it seems that a rigorous comparison between two methods can only be achieved by comparing their Pareto front. However none of the papers on LipNet1 convolutions [ 53, 54, 3, 8, 55, 56 , 57] does this to evaluate robustness. An extensive comparative study between architectures is an important future work that deserves its own 9 neurips pages.

The experiment we added in fig 16 (appendix N) shows that changing architecture size shifts the Pareto front: we expect similar behavior for other LipNet1 architectures.

---

> ### Author Response · Authors · 2022-08-01
> **General response: experiments we added**
>
> Please find below the modifications we made to the paper.
>
> We added three experiments:
> 1. We extended exemple 1 in section 5.2 to show that this phenomenon is also observed on Mnist. Results were added in fig 12. We see that the norm of the weights increases continually during training (both with SGD and Adam). This is coherent with the empirical observations made about the high Lipschitz constant of AllNet networks [23].
> 2. We now report generalization gap of AllNet networks in figure 4 for better comparison between AllNet and LipNet1.
> 3. We introduce a new experiment on Fashion-Mnist to show sensitivity of Pareto front with respect to architecture size. This experiment also shows the stability of the training with respect to Tau (fig. 15 and 16 in appendix N).
>
> We hope those additional large scale experiments can convince you that our theoretical results also hold in practice.
>
> Other modifications:
> * Better separation of results and definitions, reformatting of some equations
> * References formatting
> * Typo + minor remarks

---

### Author Response · Authors · 2022-08-06
**Available for questions and discussions**

We appreciated the thoughtful review, and are available to answer any question regarding the modifications we made.

---

### Meta-Review · Area_Chair_UDTq · 2022-08-26

**Recommendation:** Accept
**Confidence:** Certain

**Metareview:**

The submission proposes a series of novel results for Lipschitz models on robustness, generalization, and empirical performances opening a new venue for working on Lipschitz neural networks for example. While these results are important and interesting, the authors have struggled to provide a clear takeaway from this submission but discussions with reviewers have provided improvements to this paper. Despite its clarity issues and after reading the paper, I still recommend this paper for acceptance.

**Award:**

No

---

### Decision · Program_Chairs · 2022-09-14

Accept